# PROVABLY SAMPLE-EFFICIENT ROBUST REINFORCEMENT LEARNING WITH AVERAGE REWARD

## ABSTRACT

Robust reinforcement learning (RL) under the average-reward criterion is essential for long-term decision-making, particularly when the environment may differ from its specification. However, a significant gap exists in understanding the finite-sample complexity of these methods, as most existing work provides only asymptotic guarantees. This limitation hinders their principled understanding and practical deployment, especially in data-limited scenarios. We close this gap by proposing **Robust Halpern Iteration (RHI)**, a new algorithm designed for robust Markov Decision Processes (MDPs) with transition uncertainty characterized by $\ell_p$-norm and contamination models. Our approach offers three key advantages over previous methods: *(1). Weaker Structural Assumptions:* RHI only requires the underlying robust MDP to be communicating, a less restrictive condition than the commonly assumed ergodicity or irreducibility; *(2). No Prior Knowledge:* Our algorithm operates without requiring any prior knowledge of the robust MDP; *(3). State-of-the-Art Sample Complexity:* To learn an $\epsilon$-optimal robust policy, RHI achieves a sample complexity of $\tilde{\mathcal{O}}\left(\frac{SA\mathcal{H}^2}{\epsilon^2}\right)$, where $S$ and $A$ denote the numbers of states and actions, and $\mathcal{H}$ is the robust optimal bias span. This result represents the tightest known bound. Our work hence provides essential theoretical understanding of sample efficiency of robust average reward RL.

## 1 INTRODUCTION

Reinforcement Learning (RL) seeks to find an optimal policy for an agent interacting with an environment to maximize a cumulative reward. While RL has achieved remarkable success in controlled settings like board games (Silver et al., 2016; Zha et al., 2021) and video games (Wei et al., 2022; Liu et al., 2022a), its deployment in real-world applications is often hindered by a significant performance drop. This issue, known as the "Sim-to-Real" gap (Zhao et al., 2020; Peng et al., 2018; Tobin et al., 2017), stems from mismatches between the training (simulation) and deployment (real-world) environments. In contrast to games where these environments are identical, practical scenarios are fraught with model discrepancies arising from modeling errors, environmental perturbations, or even adversarial attacks (Henderson et al., 2018; Rajeswaran et al., 2016; Zhang et al., 2018). Such mismatches can render a learned policy highly suboptimal, severely undermining the reliability of RL in practice. To address this critical reliability challenge, the framework of (distributionally) robust RL was developed (Bagnell et al., 2001; Nilim & El Ghaoui, 2004; Iyengar, 2005). Instead of assuming a single, perfectly known environment model, robust RL considers an uncertainty set of plausible transition dynamics. The objective is to find a policy that optimizes performance for the worst-case model within this set. This "worst-case" approach yields a policy with formal performance guarantees across all considered environmental variations, making it inherently more resilient and robust to model mismatch and enhancing its generalizability (Pinto et al., 2017; Zhang et al., 2025).

Beyond robustness, the choice of the reward criterion fundamentally shapes the RL problem. The discounted-reward criterion, while mathematically elegant and widely studied, can be myopic due to its exponential down-weighting of future rewards, potentially leading to poor long-term outcomes (Schwartz, 1993; Seijen & Sutton, 2014; Tsitsiklis & Roy, 1997; Abounadi et al., 2001). In contrast, numerous real-world applications–such as queuing control, portfolio optimization, and communication networks (Kober et al., 2013; Lu et al., 2018; Chen et al., 2022; Wu et al., 2023; Moody & Saffell, 2001; Charpentier et al., 2021; Masoudi, 2021; Li & Hai, 2024)–demand policies that are evaluated

based on their long-term, steady-state performance when executed over an extended period of time. This practical necessity underscores the importance of the average-reward criterion, which does not discount the future reward and thus captures the long-term performance (Sigaud & Buffet, 2013). In this paper, we focus on the intersection of these two needs: developing robust RL algorithms under the average-reward criterion, to ensure performance of RL systems under model mismatch.

Robust RL under the average-reward criterion, however, is more challenging than its discounted-reward counterpart and remains relatively understudied. The primary difficulties stem from its reliance on the limiting behavior of stochastic processes, leading to analytical and algorithmic complications. Recent work has highlighted these issues, including the non-contractive nature of the associated Bellman operator, the high dimensionality of the solution space, and the instability of standard iterative algorithms (Wang et al., 2023g; Grand-Clement et al., 2023). Therefore, a critical gap in the literature persists: existing studies are predominantly asymptotic or planning based, leaving the crucial finite-sample properties of data-driven robust average-reward RL largely unexplored.

A natural strategy to obtain finite-sample results is to reduce the average-reward problem to its discounted counterpart, thereby leveraging the rich literature on robust discounted-reward RL (Wang et al., 2022; Zurek & Chen, 2023). This approach is theoretically supported by the convergence of the robust discounted value function to the average-reward value function as the discount factor approaches one (Wang et al., 2023f). However, these reduction-based methods are often suboptimal (Grand-Clément & Petrik, 2023) or require additional prior knowledge (Roch et al., 2025). While other recent works have proposed direct methods, they typically rely on strong structural assumptions, such as irreducibility, which induce a contraction property (Xu et al., 2025a;b). To circumvent these limitations, in this paper, we propose a direct approach, **Robust Halpern Iteration (RHI)**, which enables a practical, model-free implementation and achieves a near-optimal sample complexity. Our contributions are summarized as follows.

**Theoretical Foundation for Communicating Robust AMDPs.** We relax the restrictive structural assumptions common in prior work, such as irreducibility (Xu et al., 2025a) and ergodicity (Chen et al., 2025), by analyzing robust AMDPs under the weaker *communicating* condition (Bertsekas, 2011). Within this more general framework, we first establish that the optimal robust average reward is constant across all states. We then provide fundamental guarantees for the corresponding robust Bellman equation, proving its solvability and the optimality of its solution. Crucially, we formally derive the equivalence between solving this equation and finding an optimal robust policy, which provides the theoretical foundation for our algorithm's design and analysis.

**A Near-Optimal, Model-Free Algorithm for Robust Average-Reward RL.** We propose the Robust Halpern Iteration (RHI), a direct algorithm that bypasses the complexities of reduction-based approaches. Inspired by Halpern Iteration from the optimization literature (Halpern, 1967; Lieder, 2021; Lee et al., 2025), our method integrates two key technical innovations: (1) leveraging a quotient space to manage the high dimensionality of the robust Bellman equation's solution space and tackle the double unknown variables in the equation, and (2) designing a novel estimator for the robust average-reward Bellman operator. We provide a rigorous finite-sample analysis for RHI under both contamination (Wang & Zou, 2021; 2022; Jiao & Li, 2024) and $\ell_p$-norm (Kumar et al., 2023; Zhang et al., 2025) uncertainty models. Under our communicating assumption, we prove that RHI finds an $\epsilon$-optimal policy with a sample complexity of $\tilde{\mathcal{O}}\left(\frac{SA\mathcal{H}^2}{\epsilon^2}\right)$, where $S$ and $A$ are the sizes of the state and action spaces, and $\mathcal{H}$ is the span of the robust optimal bias. This result establishes the tightest near-optimal sample complexity bound for robust average-reward RL.

**Empirical Validation.** We validate the practical performance of RHI by conducting experiments across three common uncertainty models: contamination, total variation ($\ell_\infty$-norm), and $\ell_2$-norm. Our results demonstrate that RHI consistently and efficiently converges to the optimal robust average reward, computed based on the RRVI method (Wang et al., 2023g). These empirical findings corroborate our theoretical analysis and validate the convergence of RHI in practice.

## 2 PRELIMINARIES AND PROBLEM FORMULATION

**Discounted reward MDPs.** A discounted reward Markovian decision process (DMDP) $(\mathcal{S}, \mathcal{A}, \mathsf{P}, r, \gamma)$ is specified by: a state space $\mathcal{S}$, an action space $\mathcal{A}$, a nominal (stationary) transition kernel $\mathsf{P} =$

$\{P_s^a \in \Delta(\mathcal{S}), a \in \mathcal{A}, s \in \mathcal{S}\}^1$, where $P_s^a$ is the distribution of the next state over $\mathcal{S}$ upon taking action $a$ in state $s$ (with $P_{s,s'}^a$ denoting the probability of transitioning to $s'$), a reward function $r : \mathcal{S} \times \mathcal{A} \to [0, 1]$, and a discount factor $\gamma \in [0, 1)$. At each time step $t$, the agent at state $s_t$ takes an action $a_t$, the environment then transitions to the next state $s_{t+1}$ according to $P_{s_t}^{a_t}$, and produces a reward signal $r_t = r(s_t, a_t)$ to the agent.

A stationary policy $\pi : \mathcal{S} \to \Delta(\mathcal{A})$ is a distribution over $\mathcal{A}$ for any given state $s$. The agent follows the policy by taking an action following the distribution $\pi(s)$. The accumulative reward of a stationary policy $\pi$ starting from $s \in \mathcal{S}$ for DMDPs is measured by the discounted value function: $V_{\gamma,P}^\pi(s) \triangleq \mathbb{E}_{\pi,P} [\sum_{t=0}^\infty \gamma^t r_t | S_0 = s]$.

**Average reward MDPs.** Unlike DMDPs, average reward MDPs (AMDPs) do not discount the rewards over time and instead measure the accumulative reward by considering the behavior of the underlying Markov process under the steady-state distribution. Specifically, the average reward (or the gain) of a policy $\pi$ starting from $s \in \mathcal{S}$ is

$$g_P^\pi(s) \triangleq \liminf_{n \to \infty} \mathbb{E}_{\pi,P} \left[ \frac{1}{n} \sum_{t=0}^{n-1} r_t | S_0 = s \right]. \tag{1}$$

The bias or the relative value function for an AMDP is defined as the cumulative difference over time between the immediate reward and the average reward:

$$h_P^\pi(s) \triangleq \mathbb{E}_{\pi,P} \left[ \sum_{t=0}^\infty (r_t - g_P^\pi) | S_0 = s \right]. \tag{2}$$

**Distributionally robust MDPs.** In distributionally robust MDPs, the transition kernel is not fixed but, instead, belongs to a designated uncertainty set denoted as $\mathcal{P}$. Following an action, the environment undergoes a transition to the next state based on an arbitrary transition kernel $P \in \mathcal{P}$. In this paper, we mainly focus on the $(s, a)$-rectangular uncertainty set (Nilim & El Ghaoui, 2004; Iyengar, 2005; Wiesemann et al., 2013), where $\mathcal{P} = \bigotimes_{s,a} \mathcal{P}_s^a$, with $\mathcal{P}_s^a \subseteq \Delta(\mathcal{S})$ defined independently over all state-action pairs. In most studies, the uncertainty set is defined through some distribution divergence:

$$\mathcal{P}_s^a = \{q \in \Delta(\mathcal{S}) : D(q||P_s^a) \le R\}, \tag{3}$$

where $D$ is some distribution divergence like total variation, $P_s^a$ is the centroid of the uncertainty set, referred to as the nominal kernel, and $R$ is the radius of the uncertainty set for the given state and action, measuring the level of uncertainties. In most studies, the nominal kernel can be viewed as the simulation, and all training data are generated under it. In this paper, we mainly consider two widely studied models:

$$\text{Contamination model: } \mathcal{P}_s^a = \{(1 - R)P_s^a + Rq : q \in \Delta(\mathcal{S})\}, \tag{4}$$

$$\ell_p\text{-norm model: } \mathcal{P}_s^a = \{q \in \Delta(\mathcal{S}) : \|q - P_s^a\|_p \le R\}. \tag{5}$$

Robust MDPs aim to optimize the worst-case performance over the uncertainty set. With the discounted reward criterion, the robust DMDP $(\mathcal{S}, \mathcal{A}, \mathcal{P}, r, \gamma)$ consider the robust discounted value function of a policy $\pi$, which is the worst-case discounted value function over all possible transition kernels:

$$V_{\gamma,\mathcal{P}}^\pi(s) \triangleq \min_{P \in \mathcal{P}} \mathbb{E}_{\pi,P} \left[ \sum_{t=0}^\infty \gamma^t r_t | S_0 = s \right]. \tag{6}$$

The discounted robust value functions are shown to be the unique solution to the robust discounted Bellman equation (Iyengar, 2005), where $\sigma_{\mathcal{P}_s^a}(V) \triangleq \min_{P \in \mathcal{P}_s^a} PV$:

$$V(s) = \sum_a \pi(a|s)(r(s, a) + \gamma \sigma_{\mathcal{P}_s^a}(V)). \tag{7}$$

When the long-term performance under uncertainty is concerned, we focus on the robust AMDP $(\mathcal{S}, \mathcal{A}, \mathcal{P}, r)$. The worst-case performance is then measured by the following robust average reward:

$$g_\mathcal{P}^\pi(s) \triangleq \min_{P \in \mathcal{P}} \liminf_{n \to \infty} \mathbb{E}_{\pi,P} \left[ \frac{1}{n} \sum_{t=0}^{n-1} r_t | S_0 = s \right] = \min_{P \in \mathcal{P}} g_P^\pi(s). \tag{8}$$

---

[1] $\Delta(\mathcal{S})$: the $(|\mathcal{S}| - 1)$-dimensional probability simplex on $\mathcal{S}$.

The robust AMDP aims to find an optimal policy w.r.t. it: $\pi^* \triangleq \arg\max_{\pi \in \Pi} g_{\mathcal{P}}^{\pi}(s)$, for any $s \in \mathcal{S}$, and we denote the optimal robust average reward by $g_{\mathcal{P}}^* \triangleq \max_{\pi} g_{\mathcal{P}}^{\pi}$. Moreover, we define the optimal robust bias span for the robust AMDP as

$$\mathcal{H} \triangleq \max_{\mathsf{P} \in \mathcal{P}} \mathbf{Sp}(h_{\mathsf{P}}^{\pi^*}) \tag{9}$$

where $h_{\mathsf{P}}^{\pi^*}$ is the bias defined in equation 2 and $\mathbf{Sp}(h) \triangleq \max_s h(s) - \min_s h(s)$ is the Span semi-norm.

**Problem formulation.** We consider the standard *generative model setting* (Panaganti & Kalathil, 2022; Shi et al., 2023; Xu et al., 2023), where the learner assumes access to a simulator to generate i.i.d. samples under any state-action pair, following the nominal kernel P. We study the sample complexity from the nominal kernel for identifying an $\epsilon$-optimal policy $\pi$ for the robust AMDP:

$$g_{\mathcal{P}}^{\pi^*}(s) - g_{\mathcal{P}}^{\pi}(s) \leq \epsilon, \forall s \in \mathcal{S}. \tag{10}$$

## 3 COMMUNICATING RAMDPS

In this work, we consider robust AMDPs with compact uncertainty sets and satisfying the robust communicating assumption, which can be viewed as an extension of the standard weakly communicating condition in standard MDPs, e.g., (Bertsekas, 2011; Wan et al., 2021; Wan & Sutton, 2022; Zurek & Chen, 2024; 2023; Wang et al., 2022; Zhang & Xie, 2023).[2]

**Assumption 3.1.** The uncertainty set $\mathcal{P}$ is compact. Moreover, for any transition kernel $\mathsf{P} \in \mathcal{P}$, and any two states $s \neq s' \in \mathcal{S}$, there exists a stationary policy $\pi$ and some positive integer $N$, such that $\mathsf{P}^{\pi}(S_N = s'|S_0 = s) > 0$.

The robust communicating assumption assumes that for any kernel $\mathsf{P} \in \mathcal{P}$, any state $s'$ can be reached from any other state $s$ under some policy. Note that this policy may vary depending on the specific state pair and transition kernel. This condition is substantially weaker than the ergodic or irreducible assumptions made in previous robust AMDP literature Chen et al. (2025); Xu et al. (2025b;a), which require that all states inter-communicate under **any** stationary policy. It also differs from the unichain assumption (Wang et al., 2023f;g; Roch et al., 2025), which permits transient states but requires all recurrent states to form a single communicating class under **any** stationary deterministic policy. While neither our communicating assumption nor the unichain assumption strictly contains the other, however, our theoretical results can be directly applied to the unichain setting.

We then characterize structures of robust AMDPs under Assumption 3.1. Specifically, we mainly focus on the following robust Bellman equation of $(Q, g) \in \mathbb{R}^{SA} \times \mathbb{R}$:

$$Q(s,a) = r(s,a) - g + \sigma_{\mathcal{P}_s^a}(Q_{\max}), \tag{11}$$

where $\cdot_{\max} : \mathbb{R}^{SA} \to \mathbb{R}^S$ is a mapping that maps any $SA$-dimensional vector $Q$ to a $S$-dimensional vector $Q_{\max} \in \mathbb{R}^S$ with entry $Q_{\max}(s) = \max_{a \in \mathcal{A}} Q(s,a)$. This equation plays a central part in unichain robust AMDP studies, and we extend the results to our communicating setting.

**Theorem 3.2.** *Consider a robust AMDP satisfying Assumption 3.1. Then it holds that:*

*(1). The optimal robust average reward $g_{\mathcal{P}}^*$ is a constant, i.e., $g_{\mathcal{P}}^*(s_1) = g_{\mathcal{P}}^*(s_2), \forall s_1 \neq s_2$;*

*(2). The robust Bellman equation in 11 has a solution $(Q^*, g^*)$, and the solution $g^*$ is the optimal robust average reward, i.e., $g^* = g_{\mathcal{P}}^*(s)$;*

*(3). The greedy policy $\pi^*$ w.r.t. $Q$, i.e., $\pi^*(s) \in \arg\max_a Q(s,a)$, is an optimal robust policy.*

Our results extend the results for unichain robust AMDPs in (Wang et al., 2023f;g). Specifically, denote $\mathcal{T}_{\mathcal{P},g}(Q)(s,a) \triangleq r(s,a) - g + \sigma_{\mathcal{P}_s^a}(Q_{\max})$, then the robust Bellman equation equation 11 can be rewritten as $Q = \mathcal{T}_{\mathcal{P},g(Q)}$. As proved, the optimal policy $\pi^*$ can be obtained from the solution $Q^*$ to equation 11: $\pi^*(s) \in \arg\max_a Q^*(s,a)$, thus obtaining the optimal policy for our communicating robust AMDP is equivalent to solving the equation $Q = \mathcal{T}_{\mathcal{P},g_{\mathcal{P}}^*}(Q)$.

Based on this fundamental result, we develop a sample efficient algorithm to effectively solve equation 11, thus finding the optimal robust policy.

---

[2]Our communicating assumption is slightly stronger than the standard weakly communicating condition, which allows transient states to exist.

# 4 ROBUST HALPERN ITERATION (RHI) FOR ROBUST AMDPs

In this section, we design our data-driven robust Halpern Iteration (RHI) algorithm to solve equation 11. We will show later that, our RHI algorithm does not require any prior information of the robust AMDP, and achieves a near-optimal sample complexity.

As discussed in Section 2, finding the optimal policy for a robust AMDP is equivalent to solving the corresponding robust Bellman equation (11): $Q = \mathcal{T}_{\mathcal{P},g^*}(Q) = \mathcal{T}_{\mathcal{P}}(Q) - g_{\mathcal{P}}^*$, where $\mathcal{T}_{\mathcal{P}}(Q) \triangleq r + \sigma_{\mathcal{P}}(Q_{\max})$. However, solving this equation is highly challenging. Firstly, the equation has two unknown variables: $Q$ and $g_{\mathcal{P}}^*$; Since $g_{\mathcal{P}}^*$ is unknown, the operator $\mathcal{T}_{\mathcal{P},g^*}$ is not readily feasible. Moreover, different from the irreducible or ergodic cases where the operator $\mathcal{T}_{\mathcal{P},g}$ is a contraction, it is a *non-expansion* under our setting, invalidating the previous methods. Finally, the non-linear structure of $\mathcal{T}_{\mathcal{P},g}$ (compared to the linear structure of the non-robust operator) further results in a complicated solution space to the Bellman equation (Wang et al., 2023g). In the following, we address these challenges sequentially, and propose our RHI algorithm.

**Curse of dual variables.** To address the issue of solving an equation with two unknown variables, we first claim that, even if we do not know the value of $g_{\mathcal{P}}^*$, we can still obtain the optimal policy through a proximal equation. Our claim is based on the following result, where we show that a near-optimal policy can be identified by approximating the solution to the robust Bellman equation (11) w.r.t. the Span semi-norm.

**Lemma 4.1.** *Under Assumption 3.1, let $Q \in \mathbb{R}^{SA}$ and $\pi$ be the greedy policy w.r.t. $Q$, i.e., $\pi(s) \in \arg\max_{a \in \mathcal{A}} Q(s, a)$. Then, for every state $s \in \mathcal{S}$, it holds that:*

$$0 \leq g_{\mathcal{P}}^* - g_{\mathcal{P}}^{\pi}(s) \leq \boldsymbol{Sp}(\mathcal{T}_{\mathcal{P},g_{\mathcal{P}}^*}(Q) - Q) = \boldsymbol{Sp}(\mathcal{T}_{\mathcal{P}}(Q) - Q). \tag{12}$$

The result thus implies that, to obtain the optimal policy $\pi^*$, exactly solving equation 11 is not necessary; instead, it suffices to find a weaker solution $Q$ such that $\mathcal{T}_{\mathcal{P},g_{\mathcal{P}}^*}(Q) - Q = ce$, for some constant $c \in \mathbb{R}$ and the all-one vector $e = (1, ..., 1) \in \mathbb{R}^{SA}$ (note that the solution $Q$ to equation 11 also satisfies the equation with $c = 0$). Moreover, we show that this equation, and hence finding the optimal policy, are further equivalent to solving the proximal equation that only contains one variable:

$$\mathcal{T}_{\mathcal{P}}(Q) - Q = ce, \text{ for some } c \in \mathbb{R}, \tag{13}$$

since it is sufficient to find an arbitrary solution to equation 13 for some $c$. Noting that the span semi-norm is invariant to constant shifts, and inspired by previous studies of non-robust AMDPs (Zhang et al., 2021; Lee et al., 2025), we instead consider the embedded equation in the quotient space w.r.t. identical vectors. Namely, we define a relation between two vectors $v, w \in \mathbb{R}^{SA}$: $v \sim w$ if $v - w = ce$ for some $c$, which can be directly verified to be an equivalence relation. We thus construct the quotient space $E \triangleq \mathbb{R}^{SA} / \sim$, and the embedded equation of equation 13 on $E$ becomes:

$$[\mathcal{T}_{\mathcal{P}}(Q)] = [Q], \text{ where } [\cdot] \text{ denotes the equivalence class of } \cdot. \tag{14}$$

Thus, solving a robust AMDP is equivalent to solving equation 14 in the quotient space $E$. Notably, *this equation only contains one variable and has a much easier structure*.

**Non-contraction.** The second challenge is that the robust Bellman operator $\mathcal{T}_{\mathcal{P}}$ is not a contraction, but rather only a non-expansion, even in the quotient space $E$. This invalidates the previous approaches for the discounted setting or average reward setting with stronger assumptions (Chen et al., 2025; Xu et al., 2025a;b), which utilize the Banach-Picard iteration to find the unique fixed point of the contracted operator. To address this issue and find a solution to the non-expansion equation 13, we adopt the Halpern iteration (Halpern, 1967) from the stochastic approximation area. Specifically, to solve an equation $x = T(x)$ for a non-expansion operator $x$, the Halpern iteration recursively updates the algorithms through $x^{k+1} = (1 - \beta_{k+1})x^0 + \beta_{k+1}T(x^k)$, which is a convex combination between $T(x^k)$ and the initialization $x^0$. Halpern iteration has been studied in optimization areas (Halpern, 1967; Sabach & Shtern, 2017; Lieder, 2021; Park & Ryu, 2022; Contreras & Cominetti, 2023) and more recently in non-robust RL (Lee et al., 2025; Lee & Ryu, 2025).

Based on the Halpern iteration, we can similarly develop our RHI algorithm in the quotient space as $[Q^{k+1}] = [(1 - \beta_{k+1})Q^0 + \beta_{k+1}\mathcal{T}_{\mathcal{P}}(Q)]$. We show in the following result that it will converge to some solution to equation 14, and hence find an optimal policy, when the robust AMDP is known.

**Theorem 4.2.** *Consider the exact robust Halpern iteration $[Q^{k+1}] = [(1 - \beta_{k+1})Q^0 + \beta_{k+1}\mathcal{T}_{\mathcal{P}}(Q)]$, with $\beta_k = \frac{k}{k+2}$. Set $\pi^k$ to be the greedy policy w.r.t. $Q^k$. Then,*

$$\mathbf{Sp}(\mathcal{T}_{\mathcal{P}}(Q^k) - Q^k) \to 0, \text{ and } g_{\mathcal{P}}^* - g_{\mathcal{P}}^{\pi^k} \to 0, \text{ as } k \to \infty. \tag{15}$$

This result hence implies the asymptotic convergence of our RHI algorithm, even if the operator may not be a contraction. Notably, the convergence result utilizes the solvability of the robust Bellman equation, which we derived under our weaker communicating setting.

**Efficient data-driven algorithm.** The above convergence of RHI can be obtained when we exactly know the uncertainty set $\mathcal{P}$. However, in the learning setting where we do not know the worst-case kernel, we only have access to samples from the nominal kernel. This stands as the most challenging problem in the robust RL setting, since estimating the robust Bellman operator from nominal samples can be challenging, known as off-dynamic learning (Eysenbach et al., 2020; Liu & Xu, 2024; Holla, 2021). Note that the robust Bellman operator captures the dynamics under the worst-case transition kernel, which is generally different from the nominal kernel. To address this issue, a multi-level Monte-Carlo (MLMC) approach was introduced in previous works (Liu et al., 2022b; Wang et al., 2023g). However, MLMC generally results in an infinitely large sample complexity, and only guarantees asymptotic convergence, hence it cannot be applied.

To effectively estimate the robust Bellman operator while maintaining a tractable sample complexity, we propose a recursive sampling technique, inspired by (Lee et al., 2025; Jin et al., 2024b). In particular, we utilize the nominal samples to estimate the difference between two steps: $\mathcal{T}_{\mathcal{P}}(Q^k) - \mathcal{T}_{\mathcal{P}}(Q^{k-1})$. Notably, although $\mathcal{T}_{\mathcal{P}}$ is an off-dynamic term, the difference term $\mathcal{T}_{\mathcal{P}}(Q^k) - \mathcal{T}_{\mathcal{P}}(Q^{k-1})$ can be efficiently estimated under the uncertainty sets we considered, thus enabling our algorithm design. Moreover, this sampling scheme allows us to re-use the samples from previous steps, and hence improves sample efficiency. Based on this technique, we design a concrete sampling subroutine, **R-SAMPLE**, for two types of uncertainty sets: contamination model in equation 4 and $\ell_p$-norm model in equation 5. We further incorporate our R-SAMPLE sampling algorithm to propose our RHI algorithm in Algorithm 1. In our algorithm, we utilize the sampling scheme to estimate the difference between two steps, and then re-use the estimation $T^{k-1}$ of the Bellman operator for the previous step to construct the estimation $T^k$ for the current step.

---

**Algorithm 1** Robust Halpern Iteration (RHI)

---

1: **Input:** $Q^0 = 0 \in \mathbb{R}^{SA}$, $\delta \in (0, 1)$, $c_0 = 10 \cdot \ln^2(2)$, $\beta_0 = 0$
2: $\alpha = \ln(2|\mathcal{S}||\mathcal{A}|(n+1)/\delta)$
3: $T^{-1} = r; h^{-1} = 0$
4: **for** $k = 0, \ldots, n$ **do**
5: $\quad c_k = 5(k+2)\ln^2(k+2)$, $\beta_k = k/(k+2)$
6: $\quad Q^k = (1 - \beta_k)Q^0 + \beta_k T^{k-1}$
7: $\quad h^k = Q_{\max}^k$
8: $\quad m_k = \max\{\lceil \alpha c_k \mathbf{Sp}(h^k - h^{k-1})^2/\epsilon^2 \rceil, 1\}$
9: $\quad D^k = $ R-SAMPLE$(h^k, h^{k-1}, m_k)$ $\qquad$ See Appendix C for the algorithm
10: $\quad T^k = T^{k-1} + D^k$
11: **end for**
12: $\pi^n(s) \in \arg\max_{a \in A} Q^n(s, a) \quad \forall s \in S$
13: **Output:** $\pi^n$

---

We then derive the sample complexity analysis for our RHI algorithm.

**Theorem 4.3** (Performance of RHI). *Consider a robust AMDP defined by contamination or $\ell_p$-norm, satisfying Assumption 3.1 (or the unichain assumption (Wang et al., 2023g)). Set the step sizes $c_k = 5(k+2)\ln^2(k+2)$ and $\beta_k = k/(k+2)$. Then, with probability at least $1 - \delta$, the output policy $\pi^n$ is $\epsilon$-optimal:*

$$g_{\mathcal{P}}^* - g_{\mathcal{P}}^{\pi^n}(s) \le \epsilon, \tag{16}$$

*as long as the total iteration number $n$ exceeds $\frac{\mathcal{H}}{\epsilon}$, resulting in a total sample complexity of*

$$\tilde{\mathcal{O}}\left(\frac{SA\mathcal{H}^2}{\epsilon^2}\right). \tag{17}$$

Our result is the first finite sample complexity guarantee for robust AMDPs under communicating assumptions, without any prior knowledge requirement. Hence, it underscores the sample efficiency and applicability of our algorithm. Our complexity result represents the state-of-the-art in robust average reward RL (see Section 5.1 for a detailed comparison with prior works).

We note that the minimax optimal sample complexity for *non-robust* AMDPs is $\tilde{\Omega}\left(\frac{SAH}{\epsilon^2}\right)$ (Wang et al., 2022), where $H$ is the non-robust optimal span. Noting that non-robust AMDPs are special cases of robust ones, our sample complexity result matches this minimax optimal complexity in all terms except for $\mathcal{H}$, and is thus near-optimal. We also highlight that, the minimax optimal complexity for non-robust AMDPs is achievable only with prior knowledge of $H$ or other MDP parameters (Zurek & Chen, 2023; Sapronov & Yudin, 2024; Wang et al., 2023b;c); and when there is no such knowledge, non-robust algorithms also are sub-optimal (Jin et al., 2024a; Lee et al., 2025). We leave it as future research to investigate the minimax lower bound for robust AMDPs, if it is achievable without any prior knowledge, and if it can be extended to other uncertainty sets.

**Remark 4.4.** *Implementing our RHI algorithm does not require any prior knowledge, except that the total iteration number, $n$, depends on $\mathcal{H}$. Although it is common in sample complexity analysis to have an iteration number that depends on unknown underlying parameters, e.g., (Li et al., 2021a;b; Wang et al., 2024d), its concrete and practical implementations can still be challenging. To address this issue, we further modify Algorithm 1 to employ a doubling trick (Auer et al., 1995; Besson & Kaufmann, 2018; Lee et al., 2025), and propose our Parameter-Free RHI (PF-RHI) algorithm. PF-RHI is completely independent of $\mathcal{H}$, while maintaining the same sample complexity. We defer the discussion to Appendix E,*

# 5 RELATED WORK

## 5.1 COMPARISONS WITH PRIOR RESULTS

In this section, we first compare with the most related works on finite sample complexity analysis of robust average-reward RL, including (Grand-Clément & Petrik, 2024; Roch et al., 2025; Xu et al., 2025b;a; Chen et al., 2025). The comparison is summarized in Table 1.

In (Grand-Clément & Petrik, 2024; Roch et al., 2025; Chen et al., 2025), reduction-based methods are developed. In these works, a robust discounted reward RL with some specific discount factor (referred to as a reduction factor) is constructed, and its optimal robust policy is shown to be near-optimal under average reward. Thus, the sample complexity of robust average reward RL is then equivalent to that of the corresponding discounted RL with the reduction factor. In (Grand-Clément & Petrik, 2024), an upper bound on the reduction factor is derived as $\gamma \leq 1 - \frac{C}{S^S m^{S^2}}$, when the nominal kernels are rational, i.e., $\mathsf{P}_s^a = n_{s,a}/m_{s,a}$ with $n_{s,a}, m_{s,a} \in \mathbb{N}$, and $m$ is the smallest denominator among all kernel entries. However, coupling this bound with existing sample-complexity results for robust DMDPs yields exponential sample complexity for robust AMDPs. In (Chen et al., 2025), the reduction factor is set to a sample-number dependent value, and the corresponding sample complexity is derived. However, their results require stronger assumptions on the AMDP structure (uniformly ergodic) and the radius of the uncertainty set (the radius has to be small), limiting the applicability. More recently, a reduction factor $\gamma = 1 - \frac{\epsilon}{\mathcal{H}}$ is developed in (Roch et al., 2025) and sample complexity that matches ours is derived under the unichain setting. However, this reduction factor depends on the robust optimal span $\mathcal{H}$, requiring its knowledge even before learning. In practice, access to such knowledge is infeasible, and even its estimation can be challenging and inefficient (Zurek & Chen, 2023; Tuynman et al., 2024).

Another line of work (Xu et al., 2025b;a) utilizes the truncated multi-level Monte-Carlo method developed in (Wang et al., 2024b) to directly find the optimal policy. However, both works assume the underlying robust AMDP is irreducible, under which the robust Bellman operator becomes a $\gamma$-contraction w.r.t. the Span, and the sample complexity can be derived. Their method relies heavily on the contraction (which does not hold in our setting), and so cannot be applied.

Hence, compared to these prior works, our method enjoys three major advantages: (1). We require the *weakest* AMDP structure, communicating–all prior work imposes stronger structures; (2). We do not require *any* prior knowledge of the robust AMDP (like $\mathcal{H}$ in (Roch et al., 2025)); (3). We enjoy the tightest sample complexity (noting that $\mathcal{H} \leq t_m$, i.e., the mixing time (Wang et al., 2022; Roch et al., 2025)). Thus, our RHI method represents the state-of-the-art in robust average reward RL.

| Algorithm | AMDP Structure | Uncertainty Set | Sample Complexity |
|---|---|---|---|
| (Grand-Clément & Petrik, 2024) | $P \in \mathbb{Q}$ | N/A | Exponential |
| Chen et al. (2025) | Uniformly ergodic | KL | $\tilde{\mathcal{O}}\left(\frac{SAt_m^2}{p_\wedge \epsilon^2}\right)$ |
| Xu et al. (2025b) | Irreducible & aperiodic | TV | $\tilde{\mathcal{O}}\left(\frac{SAt_m^2}{(1-\gamma)^2\epsilon^2}\right)$ |
| Xu et al. (2025a) | Irreducible & aperiodic | TV | $\tilde{\mathcal{O}}\left(\frac{SAt_m^2}{(1-\gamma)^2\epsilon^2}\right)$ |
| Roch et al. (2025) | Unichain | TV | $\tilde{\mathcal{O}}\left(\frac{SA\mathcal{H}^2}{\epsilon^2}\right)$ |
| Ours | Communicating/unichain | $l_p$ | $\tilde{\mathcal{O}}\left(\frac{SA\mathcal{H}^2}{\epsilon^2}\right)$ |

Table 1: Comparison with prior results. $t_m$ denotes the robust mixing time; $\gamma$ in (Xu et al., 2025a;b) is the contraction coefficient under the irreducibility assumption.

## 5.2 OTHER RELATED WORK

**Robust RL with average reward.** Studies on robust RL with average reward are relatively limited. Early research focused on dynamic programming (DP) methods in robust AMDPs. These investigations, initiated by (Tewari & Bartlett, 2007) for specific finite-interval uncertainty sets, were subsequently extended to more general uncertainty models in works such as (Wang et al., 2023f; Grand-Clement et al., 2023; Wang & Si, 2025). These foundational studies were instrumental in revealing the fundamental structure of robust AMDPs and illustrating their connections to robust DMDPs. As an alternative method, (Chatterjee et al., 2023) recently proposed a game-theoretic approach for finding the optimal policy. Building on the understanding of robust AMDP structures, the focus also extends to learning algorithms, where (Wang et al., 2023g) introduced a model-free algorithm with asymptotic convergence guarantees. However, all of these aforementioned approaches focus on asymptotic convergence only, leaving finite-sample complexity analyses largely unaddressed.

**Robust RL with discounted rewards.** Robust DMDPs were first studied in foundational works such as (Iyengar, 2005; Nilim & El Ghaoui, 2004; Bagnell et al., 2001; Wiesemann et al., 2013; Lim et al., 2013). These initial investigations typically assumed a fully known uncertainty set and developed solutions based on robust DP. Since then, extensive theoretical research has significantly adapted and extended these concepts to various learning paradigms where the uncertainty set or the nominal model might be unknown or learned from data. Prominent research directions include analyses in settings with generative models (Yang et al., 2022; Panaganti & Kalathil, 2022; Xu et al., 2023; Shi et al., 2023; Zhou et al., 2021; Wang et al., 2023d; Liang et al., 2023; Liu et al., 2022b; Wang et al., 2023e; 2024b; 2023a; Kumar et al., 2023; Derman et al., 2021), investigations into offline learning from fixed datasets (Shi & Chi, 2022; Liu & Xu, 2024; Wang et al., 2024a;c), and developments within online learning frameworks involving exploration (Wang & Zou, 2021; Lu et al., 2024; Ghosh et al., 2025; He et al., 2025). A key focus across these diverse settings is often to provide rigorous finite-sample complexity guarantees or convergence rates, characterized under different assumptions regarding the structure of the uncertainty set and the nature of data access.

**Non-robust RL with average reward.** The study of non-robust AMDPs originated with foundational model-based DP techniques, such as Policy Iteration and Value Iteration, which assume a known model (Puterman, 2014; Bertsekas, 2011). Subsequently, research shifted towards model-free RL algorithms. These include adaptations of Q-learning and SARSA, like RVI Q-learning (Abounadi et al., 2001; Wan et al., 2021; Wan & Sutton, 2022), designed to learn optimal policies directly from interaction data without requiring explicit model knowledge (Dewanto et al., 2020).

Beyond asymptotic convergence, sample complexity for achieving near-optimal policies in (non-robust) AMDPs is extensively studied. A significant body of work is based on the reduction framework, which transforms the AMDP into a DMDP using a carefully chosen discount factor. However, selecting an appropriate discount factor typically requires prior knowledge of crucial MDP parameters, such as the span of the bias function (Zurek & Chen, 2023; Wang et al., 2022; Zurek & Chen, 2024; Sapronov & Yudin, 2024; Jin & Sidford, 2021) or various mixing time constants (Wang et al., 2023b;c). Notable progress has been made under such assumptions, for instance, (Zurek & Chen, 2023; Sapronov & Yudin, 2024) demonstrate that if the bias span is known and used to set the

reduction factor, the resulting sample complexity matches the minimax optimal rate for weakly communicating MDPs (Wang et al., 2022). Alongside reduction-based methods, direct approaches that do not involve conversion to DMDPs, but still require prior knowledge, have also been recently developed (Zhang & Xie, 2023; Li et al., 2024). Recognizing that the prerequisite of prior knowledge can be restrictive and impractical, and that estimating these parameters accurately is challenging (Tuynman et al., 2024), another line of research investigates AMDPs without prior knowledge, achieving sub-optimal sample complexity (Lee et al., 2025; Jin et al., 2024a; Lee & Ryu, 2025; Tuynman et al., 2024).

Extending these diverse frameworks and insights to robust AMDPs is, however, particularly challenging. This difficulty stems from the greater complexity inherent in the robust average-reward paradigm, including issues such as the non-linearity of the robust Bellman operator and a more intricate, high-dimensional solution space for the robust Bellman equation (Wang et al., 2023g).

## 6 EXPERIMENT RESULTS

We conduct experiments to validate our theoretical results and evaluate the empirical performance of RHI. We consider the Garnet problem (Archibald et al., 1995) G(20,15) with 20 states and 15 actions, where nominal transition kernels are randomly generated. We consider three uncertainty sets: the contamination model, the $\ell_\infty$-norm model (total variation), and the $\ell_2$-norm model.

After each iteration of our RHI algorithm, we derive the greedy policy based on the current Q-value estimates from RHI. The robust average reward of this derived policy is then calculated using the RRVI algorithm from (Wang et al., 2023g) and recorded. For comparison, we establish a baseline consisting of the optimal robust average reward, also computed via the RRVI algorithm. Each experimental configuration is repeated for 10 independent runs. All of our experiments require minimal compute resources and are implemented using Google Colab. We present the mean robust average reward across these runs where the shaded region in Figure 1 is the standard deviation.

As depicted in Figure 1, the experimental results demonstrate that our RHI algorithm effectively converges to the optimal robust average reward, thereby corroborating our theoretical findings.

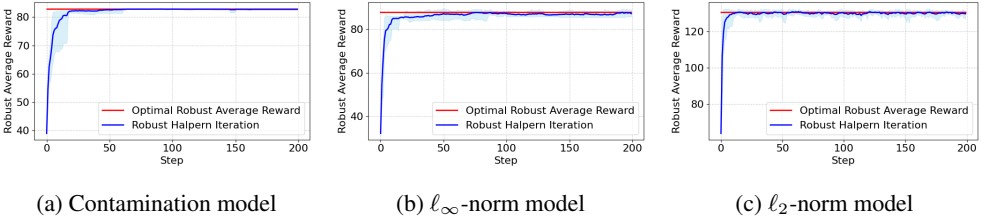

(a) Contamination model        (b) $\ell_\infty$-norm model        (c) $\ell_2$-norm model

Figure 1: Performance of RHI.

## 7 CONCLUSION

Robust reinforcement learning under the average-reward criterion suffers from the significant challenge of developing efficient algorithms with finite-sample guarantees, thus hindering its application in data-limited environments. This generally resulted from complexity of the problem setting and the limitations of prior approaches, which often relied on stronger structural assumptions, or required impractical prior knowledge. Therefore, we introduced Robust Halpern Iteration (RHI), a novel model-free algorithm for finding near-optimal policies in robust AMDPs. Key advantages of RHI are its ability to bypass the need for prior knowledge of specific MDP parameters or strong AMDP structures, which are common prerequisites for prior methods. We theoretically established that RHI achieves a sample complexity of $\tilde{\mathcal{O}}\left(\frac{SA\mathcal{H}^2}{\epsilon^2}\right)$ to find an $\epsilon$-optimal policy, under the contamination/unichain conditions and $\ell_p$-norm/contamination uncertainty sets. Our result is near-optimal, enhancing the applicability of average-reward robust RL in those data-intensive and real-world applications.

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

# A  PRELIMINARIES AND PROOF ORGANIZATION

To facilitate the analysis and understanding of our work, we specify the notations as follows.

**System Characteristics.** We consider a robust AMDP $(\mathcal{S}, \mathcal{A}, \mathcal{P}, r)$ centered around the nominal kernel P with the following properties:

- The uncertainty set $\mathcal{P} = \bigotimes_{s \in \mathcal{S}, a \in \mathcal{A}} \mathcal{P}_s^a$ is $SA$-rectangular (Nilim & El Ghaoui, 2004; Iyengar, 2005; Wiesemann et al., 2013), where $\mathcal{P}_s^a \subseteq \Delta(\mathcal{S})$ is defined independently $\forall (s, a) \in \mathcal{S} \times \mathcal{A}$.

- Each $\mathcal{P}_s^a$ is simultaneously compact and convex.

- The robust system is communicating, meaning that for any arbitrary transition kernel $\mathsf{P} \in \mathcal{P}$ and $s_1, s_2 \in \mathcal{S}$ s.t. $s_1 \neq s_2$, there exists some stationary policy $\pi$ and integer $N$ s.t. $\mathsf{P}^\pi(S_N = s_2 | S_0 = s_1) > 0$.

- The learner in the robust system has access to a generative model or simulator (Panaganti & Kalathil, 2022; Shi et al., 2023; Xu et al., 2023) to generate i.i.d. samples for any state-action pair under the nominal kernel P.

**Additional notation.**

- We define a stationary policy as $\pi : \mathcal{S} \to \Delta(\mathcal{A})$, and subsequently define the finite set of all stationary policies as $\Pi$ such that $\pi \in \Pi$.

- Since the worst-case robust average reward under the time varying model is equivalent to the one under the stationary model (Wang et al., 2023f), we therefore focus on this time invariant model. For a given stationary policy, $\pi \in \Pi$, satisfying equation 8, we define the set of minimizing (worst-case) transition kernels as $\Omega_g^\pi \triangleq \{\mathsf{P} \in \mathcal{P} : g_\mathsf{P}^\pi = g_\mathcal{P}^\pi\}$, where $g_\mathsf{P}^\pi(s) \triangleq \liminf_{T \to \infty} \mathbb{E}_{\pi, \mathsf{P}} \left[ \frac{1}{T} \sum_{n=0}^{T-1} r_t | S_0 = s \right]$.

- We use $r$ to denote the $SA$-dimensional vector, whose $(s, a)$-th entry is $r(s, a)$. We use $\mathsf{P}_{s,s'}^a$ to denote the transition probability from $s$ to $s'$ under the action $a$ of some transition kernel P.

- Given a policy $\pi$, a reward $r$ and a transition kernel P, we denote the induced reward and state-transition kernel by $r_\pi \in \mathbb{R}^S$ and $\mathsf{P}_\pi \in \mathbb{R}^{S \times S}$:

$$r^\pi(s) = \sum_a \pi(a|s) r(s, a), (\mathsf{P}^\pi)_{s,s'} = \sum_a \pi(a|s) \mathsf{P}_{s,s'}^a. \tag{18}$$

- For a vector $V \in \mathbb{R}^S$, we use $\mathsf{P}V$ to denote an $SA$-dimensional vector as

$$(\mathsf{P}V)_{s,a} = \mathsf{P}_s^a V. \tag{19}$$

Specifically, for $Q \in \mathbb{R}^{SA}$, $Q_{\max} \in \mathbb{R}^S$, and

$$(\mathsf{P}(Q_{\max}))_{s,a} = \mathsf{P}_s^a(Q_{\max}) = \sum_{s'} \mathsf{P}_{s,s'}^a \max_b \{Q(s', b)\}. \tag{20}$$

- For an uncertainty set $\mathcal{P}$, we denote the robust Bellman operator $\mathcal{T}_\mathcal{P}(Q) : \mathbb{R}^{SA} \to \mathbb{R}^{SA}$ as

$$\mathcal{T}_\mathcal{P}(Q)(s, a) = r(s, a) + \sigma_{\mathcal{P}_s^a}(Q_{\max}). \tag{21}$$

# B  PROOF OF THEOREM 3.2

**Theorem B.1.** *(Restatement of Theorem 3.2) Consider a robust AMDP satisfying Assumption 3.1. Then it holds that:*

*(1). The optimal robust average reward $g_\mathcal{P}^*$ is a constant, i.e., $g_\mathcal{P}^*(s_1) = g_\mathcal{P}^*(s_2), \forall s_1 \neq s_2$;*

*(2). The robust Bellman equation in 11 has a solution $(Q^*, g^*)$, and the solution $g^*$ is the optimal robust average reward, i.e., $g^* = g_\mathcal{P}^*(s)$;*

*(3). The greedy policy $\pi$ w.r.t. $Q$, i.e., $\pi(s) \in \arg\max_a Q(s, a)$, is an optimal robust policy.*

*Proof.* **Proof of (1).** Note that Assumption 3.1 implies that for any $\mathsf{P} \in \mathcal{P}$, the non-robust MDP $(\mathcal{S}, \mathcal{A}, \mathsf{P}, r)$ is weakly accessibility (Zurek & Chen, 2023), thus the optimal average reward $g_\mathsf{P}^*$ is a constant (Bertsekas, 2011).

We then apply Theorem 3.5 from (Grand-Clement et al., 2023), which shows that the optimal robust average gain, $g_\mathcal{P}^*$, is the value of the zero-sum stochastic game between the agent and the environment, and the following saddle-point equilibrium exists:

$$g_\mathcal{P}^* = \sup_\pi \inf_{\mathsf{P} \in \mathcal{P}} g_\mathsf{P}^\pi = \inf_{\mathsf{P} \in \mathcal{P}} \sup_\pi g_\mathsf{P}^\pi. \tag{22}$$

Since for a fixed $\mathsf{P}$, $\sup_\pi g_\mathsf{P}^\pi = g_\mathsf{P}^*$ is a constant, thus the RHS of equation 22 is also a constant, as the infimum over a set of scalar constants is itself a scalar constant. This hence proves that the optimal robust gain $g_\mathcal{P}^*$ is a constant, independent of the initial state.

**Proof of (2).** As $g_\mathcal{P}^*$ is a constant under our setting, it satisfies the initial-state-independent condition in (Wang & Si, 2025), thus part (2) can be directly obtained by applying the results in (Wang & Si, 2025).

**Proof of (3).** Since $(Q^*, g^*)$ is a solution to the robust Bellman equation equation 11, the pair $(h^*, g^*)$, where $h^*(s) \triangleq \max_a Q^*(s, a)$ satisfies the following equation:

$$h^*(s) + g^* = \sum_a \pi(a|s)(r(s,a) + \sigma_{\mathcal{P}_s^a}(h^*)).$$

Let $\mathsf{P}_\pi \in \mathcal{P}$ be the worst-case transition kernel for policy $\pi$. Then it holds that

$$h^*(s) + g^* = \sum_a \pi(a|s)(r(s,a) + \sigma_{\mathcal{P}_s^a}(h^*)) = \sum_a \pi(a|s)(r(s,a) + (\mathsf{P}_\pi)_s^a(h^*)),$$

i.e.,

$$h^* = r_\pi - g^* + (\mathsf{P}_\pi)^\pi h^*. \tag{23}$$

Multiplying this inequality by $((\mathsf{P}_\pi)^\pi)^k$ and taking a sum further implies that

$$g^* = \frac{\sum_{k=0}^{n-1}((\mathsf{P}_\pi)^\pi)^k r_\pi}{n} + \frac{(((\mathsf{P}_\pi)^\pi)^n - I)h^*}{n}. \tag{24}$$

We then take $\liminf$ on both sides, and it implies that

$$g^* = \liminf_{n \to \infty} \frac{\sum_{k=0}^{n-1}((\mathsf{P}_\pi)^\pi)^k r_\pi}{n} = g_\mathcal{P}^\pi, \tag{25}$$

since $(\mathsf{P}^k - I)h^*$ is bounded and finite. Since (2) implies that $g^* = g_\mathcal{P}^*$, thus $g_\mathcal{P}^\pi = g_\mathcal{P}^*$, and the greedy policy $\pi$ is optimal.

We hence complete the proof. $\qquad\square$

## C  SAMPLING ALGORITHM

In this section, we present a method to approximate the robust Bellman operator $T^k \approx \mathcal{T}_\mathcal{P}(Q^k)$ by sampling from the nominal kernel $\mathsf{P}$. Our method is based on the concrete closed-form of the support function $\sigma_\mathcal{P}(\cdot)$ over the two considered uncertainty sets.

$\ell_p$**-norm sets.** When the uncertainty set is defined through the $\ell_p$-norm as in equation 5, it is shown that the robust Bellman operator has the following closed-form solution in (Kumar et al., 2023):

$$\mathcal{T}_\mathcal{P}(Q^k) = r + \mathsf{P}(Q_{\max}^k) - R\kappa(Q_{\max}^k), \tag{26}$$

with some penalty function $\kappa$ that is independent from $\mathsf{P}$. We defer the constructions of $\kappa$ to Remark D.2.

**Contamination set.** With contamination set in equation 4, it holds that (Wang & Zou, 2021):

$$\mathcal{T}_\mathcal{P}(Q^k) = r + (1-R)\mathsf{P}(Q_{\max}^k) + R \min_s(Q_{\max}^k). \tag{27}$$

Note that for both uncertainty sets, the difference $\mathcal{T}_{\mathcal{P}}(Q_1) - \mathcal{T}_{\mathcal{P}}(Q_2)$ further can be derived, which facilitates our estimation. To re-use the pre-collected samples to enhance sample efficiency, we further develop our difference-based algorithm.

Specifically, for the $\ell_p$-norm case, let $h^k = Q_{\max}^k$ and $h^{k-1} = Q_{\max}^{k-1}$, and we set the difference terms $d^k = h^k - h^{k-1}$, and $K^k = \kappa(h^k) - \kappa(h^{k-1})$. Then it holds that

$$\mathcal{T}_{\mathcal{P}}(Q^k) - \mathcal{T}_{\mathcal{P}}(Q^{k-1}) = \mathsf{P}d^k + K^k. \tag{28}$$

Hence it suffices to estimate $\mathsf{P}d^k$ in our algorithm. We present our robust sampling algorithm (R-SAMPLE) as follows.

---

**Algorithm 2** R-SAMPLE$(h^k, h^{k-1}, m)$

---

1: **Input:** $h^k$, $h^{k-1}$, $m$
2: **for** $(s,a) \in \mathcal{S} \times \mathcal{A}$ **do**
3:      **if** $\ell_p$-norm uncertainty set **then**
4:          Compute $d^k = h^k - h^{k-1}$ and $K^k = \kappa(h^k) - \kappa(h^{k-1})$
5:          $D^k(s,a) = \frac{1}{m}\sum_{j=1}^m d^k(s_j) - RK^k(s,a)$ with $s_j \overset{\text{iid}}{\sim} \mathsf{P}_s^a$
6:      **end if**
7:      **if** Contamination uncertainty set **then**
8:          Compute $d^k = h^k - h^{k-1}$ and $K^k = \min_s(h^k) - \min_s(h^{k-1})$
9:          $D^k(s,a) = \frac{1-R}{m}\sum_{j=1}^m d^k(s_j) + RK^k(s,a)$ with $s_j \overset{\text{iid}}{\sim} \mathsf{P}_s^a$
10:      **end if**
11: **end for**
12: **Output:** $D^k$

---

## D    PROOFS FOR RHI

### D.1    ANALYSIS OF RHI

**Lemma D.1** (Restatement of Lemma 4.1). *Under Assumption 3.1, let $Q \in \mathbb{R}^{SA}$ and $\pi$ be the greedy policy w.r.t. $Q$, i.e., $\pi(s) \in \arg\max_{a \in \mathcal{A}} Q(s,a)$. Then for every state $s \in \mathcal{S}$, it holds that:*

$$0 \le g_{\mathcal{P}}^* - g_{\mathcal{P}}^\pi(s) \le \mathbf{Sp}(\mathcal{T}_{\mathcal{P}, g_{\mathcal{P}}^*}(Q) - Q) = \mathbf{Sp}(\mathcal{T}_{\mathcal{P}}(Q) - Q).$$

*Proof.* Denote $h(s) \triangleq Q_{\max}(s) = \max_a Q(s,a)$. Since $\pi$ is greedy w.r.t $Q$, it follows that $h(s) = Q(s, \pi(s))$ for all $s \in \mathcal{S}$. We first denote the worst-case transition kernel of $h$ over $\mathcal{P}$ by $\mathsf{P}$, and its induced kernel by $\mathsf{P}_\pi$, i.e.,

$$(\mathsf{P}_\pi h)(s) = \min_{\mathsf{P} \in \mathcal{P}_s^{\pi(s)}} \mathbb{E}_{s' \sim \mathsf{P}}[h(s')] = \sigma_{\mathcal{P}_s^{\pi(s)}}(h), \quad \forall s \in \mathcal{S}. \tag{29}$$

The robust average reward under Assumption 3.1, $g_{\mathcal{P}}^\pi$, exists and is the average reward under the worst-case kernel $\mathsf{P}_\pi$, thus it holds that

$$g_{\mathcal{P}}^\pi = g_{\mathsf{P}_\pi}^\pi = \mathsf{P}_\pi^\infty r_\pi, \tag{30}$$

where $r_\pi = r(s, \pi(s))$ and $\mathsf{P}_\pi^\infty$ is the Cesaro limit of $\mathsf{P}_\pi$ (Puterman, 2014). Note that it holds that $\mathsf{P}_\pi^\infty = \mathsf{P}_\pi^\infty \mathsf{P}_\pi$ (Puterman, 2014), thus applying this fact to equation 30 yields that

$$g_{\mathcal{P}}^\pi = \mathsf{P}_\pi^\infty (r_\pi + \mathsf{P}_\pi h - h). \tag{31}$$

We further note that the $(s', \pi(s'))$-th entry of $(\mathcal{T}_{\mathcal{P}}(Q) - Q)$ is in fact $(r_\pi(s') + (\mathsf{P}_\pi h)(s') - h(s'))$, thus it holds that

$$\min_{s' \in \mathcal{S}, a' \in \mathcal{A}} (\mathcal{T}_{\mathcal{P}}(Q) - Q)(s', a') \le (\mathcal{T}_{\mathcal{P}}(Q) - Q)(s', \pi(s)) = (r_\pi(s) + (\mathsf{P}_\pi h)(s') - h(s')),$$

and by multiplying by $\mathsf{P}_\pi^\infty$ recursively and equation 31 we have that

$$\min_{s'\in\mathcal{S},a'\in\mathcal{A}}(\mathcal{T}_{\mathcal{P}_\pi}(Q)-Q)(s',a') \le \mathsf{P}_\pi^\infty\left(r_\pi + (\mathsf{P}_\pi h) - h\right) = g_{\mathcal{P}}^\pi. \tag{32}$$

On the other hand, denote the optimal robust policy as $\pi^*$ and its associated optimal average reward as $g_{\mathcal{P}}^*$. Let $\mathsf{P}_{\pi^*} \in \mathcal{P}$ be the corresponding worst-case transition kernel. Similar to equations 30-31, we have that,

$$g_{\mathcal{P}}^* = g_{\mathsf{P}_{\pi^*}}^{\pi^*} = \mathsf{P}_{\pi^*}^\infty r_{\pi^*} = \mathsf{P}_{\pi^*}^\infty (r_{\pi^*} + \mathsf{P}_{\pi^*} h - h), \tag{33}$$

We introduce an auxiliary function $h' \in \mathbb{R}^S$ as $h'(s') \triangleq Q(s',\pi^*(s'))$ for all $s' \in \mathcal{S}$. By definition of $h$ and $h'$, we have $h'(s') \le h(s')$ which implies that $-h(s') \le -h'(s')$. Substituting this in equation 33 implies that for all $s \in \mathcal{S}$,

$$\begin{aligned}
g_{\mathcal{P}}^*(s) &= \mathsf{P}_{\pi^*}^\infty(r_{\pi^*}(s') + (\mathsf{P}_{\pi^*}h)(s') - h(s')) \\
&\le \mathsf{P}_{\pi^*}^\infty(r_{\pi^*}(s') + (\mathsf{P}_{\pi^*}h)(s') - h'(s')).
\end{aligned} \tag{34}$$

Now we note that $(r_{\pi^*}(s') + (\mathsf{P}_{\pi^*}h)(s') - h'(s'))$ is exactly the $(s',\pi^*(s'))$-th entry of $(\mathcal{T}_{\mathcal{P}}(Q)-Q)$, then it holds that

$$\begin{aligned}
g_{\mathcal{P}}^*(s) &= \mathsf{P}_{\pi^*}^\infty(r_{\pi^*}(s') + (\mathsf{P}_{\pi^*}h)(s') - h(s')) \\
&\le \mathsf{P}_{\pi^*}^\infty(r_{\pi^*}(s') + (\mathsf{P}_{\pi^*}h)(s') - h'(s')) \\
&\le \mathsf{P}_{\pi^*}^\infty \cdot \max_{s'\in\mathcal{S},a'\in\mathcal{A}}(r(s',a') + (\mathsf{P}_{\pi^*}h)(s') - h'(s')) \\
&= \max_{s',a'}(\mathcal{T}_{\mathcal{P}}(Q)-Q)(s',a')
\end{aligned} \tag{35}$$

Thus together with equation 32, it implies that

$$g_{\mathcal{P}}^* - g_{\mathcal{P}}^\pi(s) \le \max_{s',a'}(\mathcal{T}_{\mathcal{P}}(Q)-Q)(s',a') - \min_{s',a'}(\mathcal{T}_{\mathcal{P}}(Q)-Q)(s',a') = \mathbf{Sp}(\mathcal{T}_{\mathcal{P}}(Q)-Q). \tag{36}$$

It hence completes the proof by noting that $\mathbf{Sp}(\mathcal{T}_{\mathcal{P},g_{\mathcal{P}}^*}(Q)-Q) = \mathbf{Sp}(\mathcal{T}_{\mathcal{P}}(Q)-Q)$ since $g_{\mathcal{P}}^*$ is a constant per Theorem 3.2. $\qquad\square$

**Remark D.2.** *Let $F_{s,a} \subset \mathcal{S}$ be a subset of forbidden states, namely when the system is at state $s \in \mathcal{S}$ and taking action $a \in \mathcal{A}$, it is unfeasible for the system to transition to certain other states. Formally, by denoting the nominal kernel as $\tilde{\mathsf{P}}$ we have*

$$\tilde{\mathsf{P}}(s'|s,a) = \mathsf{P}(s'|s,a) = 0, \quad \forall \mathsf{P} \in \mathcal{P}, \forall s' \in F_{s,a}.$$

*We can then rewrite our kernel noise in equation 3 as*

$$\mathcal{P}_s^a = \left\{ \mathsf{P}\,\middle|\, \|\mathsf{P}\|_p = R, \sum_{s'}\mathsf{P}(s') = 0, \mathsf{P}(s'') = 0, \forall s'' \in F_{s,a} \right\}$$

*Under consideration of the $\ell_p$-norm model in equation 5 it can be shown*

$$\begin{aligned}
\kappa(h,s,a) &= \min_{\|\mathsf{P}\|_p = R, \sum_{s'}\mathsf{P}(s')=0, \mathsf{P}(s'')=0, \forall s'' \in F_{s,a}} \langle \mathsf{P}, h \rangle \\
&= \min_{\omega \in \mathbb{R}} \|u - \omega\mathbf{1}\|_p, \quad \text{where } u(s) = h(s)\mathbf{1}(s \notin F_{s,a}), \\
&= \kappa_p(u).
\end{aligned}$$

*For a concrete example within the context of our empirical results for the $\ell_\infty$ (total variation) model in Figure 1b, we have*

$$\kappa_\infty(h,s,a) = \frac{\max_{s\notin F_{s,a}} h(s) - \min_{s\notin F_{s,a}} h(s)}{2}.$$

*This construction of $\kappa$ is what allows us to directly apply Theorem 8 from (Kumar et al., 2023) by considering the $\ell_p$-ball of transition kernels $\|\tilde{\mathsf{P}} - \mathsf{P}\|_p \le R$ for all $\mathsf{P} \in \mathcal{P}$ and the nominal kernel $\tilde{\mathsf{P}}$ as turning into a penalty on the next state's value function. Adding this penalty during sampling in Algorithm 2 allows us to effectively sample from the worst-case kernel with only access to the nominal environment.*

We then present the proofs for our Robust Halpern Iteration for $\ell_p$-normed Robust AMDPs. The proof for contamination models can be derived similarly and is hence omitted.

**Proposition D.3.** *Let $c_k > 0$ with $2\sum_{k=0}^{\infty} c_k^{-1} \leq 1$ and $T^k, Q^k$ the iterates generated by $RHI(Q^0, n, \epsilon, \delta)$. Then, with probability at least $1 - \delta$ we have that $\|T^k - \mathcal{T}_{\mathcal{P}}(Q^k)\|_{\infty} \leq \epsilon$ simultaneously for all $k = 0, 1, \ldots, n$.*

*Proof.* We fix an $(s, a)$-pair in our analysis, denote $Y^i \triangleq D^i - \mathsf{P}d^i - K^i$ and $X^k \triangleq \sum_{i=0}^{k} Y^i$. Recall that $d^i = h^i - h^{i-1}$, then it holds that for all $(s, a) \in \mathcal{S} \times \mathcal{A}$ and any $i$,

$$\sigma_{\mathcal{P}_s^a}(h^i) - \sigma_{\mathcal{P}_s^a}(h^{i-1}) = \mathsf{P}d^i - R\kappa(h^i) + R\kappa(h^{i-1}), \tag{37}$$

where the $R\kappa(\cdot)$ is the penalty term from (Kumar et al., 2023), which we discuss further in Remark D.2.

Since $h^{-1} = 0$ by the initialization of RHI, we have the robust Bellman operator as

$$\mathcal{T}_{\mathcal{P}}(Q^k)(s, a) = r(s, a) + \sigma_{\mathcal{P}_s^a}(h^k) = r(s, a) - R\kappa(h^k, s, a) + \sum_{s' \in \mathcal{S}} \mathsf{P}(s'|s, a)h(s'). \tag{38}$$

We further denote that $K^i \triangleq R\kappa(h^i) - R\kappa(h^{i-1})$, and from equation 28 we have that

$$\mathcal{T}_{\mathcal{P}}(Q^k) = r + \sum_{i=0}^{k} K^i. \tag{39}$$

We then consider the estimation error. Recall that $T^k(s, a) = T^{k-1}(s, a) + D^k(s, a)$, thus

$$\begin{aligned} T^k(s, a) &- \mathcal{T}_{\mathcal{P}_s^a}(Q^k)(s, a) \\ &= T^{k-1}(s, a) - \mathcal{T}_{\mathcal{P}_s^a}(Q^{k-1})(s, a) + D^k(s, a) - \mathsf{P}d^k - K^k \\ &= T^{k-1}(s, a) - \mathcal{T}_{\mathcal{P}_s^a}(Q^{k-1})(s, a) + Y^k(s, a) \\ &= X^k(s, a), \end{aligned} \tag{40}$$

due to our initialization.

We then estimate $\mathbb{P}(\|X^k(s, a)\|_{\infty} \geq \epsilon), \forall(s, a)$ by adapting the arguments of the Azuma-Hoeffding inequality as in (Lee et al., 2025). We consider the filtration $\mathcal{F}^k = \sigma(\{D^i\}_{i=0}^k)$. Since $h^k$, $d^k$, and $m_k$ are $\mathcal{F}_{k-1}$-measurable and the relation between the robust and non-robust Bellman operators (Kumar et al., 2023), it follows that $\mathbb{E}[Y^k(s, a)|\mathcal{F}_{k-1}] = 0$ for all $(s, a)$ during sampling. Thus the sequence $\{X^k(s, a)\}_{k \geq 0}$ is a $\mathcal{F}^k$-martingale. Using Markov's inequality and the tower property of conditional expectation yields that for every $(s, a) \in \mathcal{S} \times \mathcal{A}$ and $\lambda > 0$,

$$\begin{aligned} \mathbb{P}(X^k(s, a) \geq \epsilon) &\leq e^{-\lambda\epsilon}\mathbb{E}[\exp(\lambda X^k(s, a))] \\ &= e^{-\lambda\epsilon}\mathbb{E}\big[\exp(\lambda X^{k-1}(s, a))\mathbb{E}[\exp(\lambda Y^k(s, a))|\mathcal{F}^{k-1}]\big]. \end{aligned} \tag{41}$$

Moreover, since $K^i$ is deterministic and independent from $\mathsf{P}$, it holds that $Y^k = \frac{1}{m_k}\left(\sum_j^{m_k} d^k(s_{k,j}^{s,a})\right) - \mathsf{P}d^k(s, a)$. Now since $d^k(s_{k,j}^{s,a}) \in [\min_{s'} d^k(s'), \max_{s'} d^k(s')]$ and $\mathbb{E}[Y^k(s, a)|\mathcal{F}_{k-1}] = 0$, Hoeffding's inequality yields that

$$\mathbb{E}[\exp(\lambda Y^k(s, a))|\mathcal{F}_{k-1}] = \prod_{j=1}^{m_k} \mathbb{E}\left[\exp(\lambda Y_j^k)|\mathcal{F}^{k-1}\right]$$

$$\leq \exp\left(\frac{1}{2}\lambda^2 \mathbf{Sp}(d^k)^2/m_k\right), \tag{42}$$

where $Y_j^k = \frac{1}{m_k}\left(d^k(s_{k,j}^{s,a})\right) - \frac{1}{m_k}\mathsf{P}d^k(s, a)$, and the last inequality is due to the fact that $|Y_j^k| \leq \frac{\mathbf{Sp}(d^k)}{m_k}$.

Combining equation 41 and equation 42 along with $m_k \geq \alpha c_k \mathbf{Sp}(d^k)^2/\epsilon^2$, it can be derived that

$$\mathbb{E}[\exp(\lambda X^k(s, a))] \leq \exp\left(\frac{1}{2}\lambda^2\epsilon^2 \sum_{i=0}^{k} c_i^{-1}/\alpha\right). \tag{43}$$

Combining this with $\sum_{i=0}^{\infty} c_i^{-1} \le \frac{1}{2}$, we have $\mathbb{P}(X^k(s,a) \ge \epsilon) \le \exp(-\lambda\epsilon + \frac{1}{4}\lambda^2\epsilon^2/\alpha)$. Taking $\lambda = 2\alpha/\epsilon$ we can obtain

$$\mathbb{P}(X^k(s,a) \ge \epsilon) \le \exp(-\alpha) = \frac{\delta}{2|\mathcal{S}||\mathcal{A}|(n+1)}.$$

Synonymously, we can find the same bound for $\mathbb{P}(X^k(s,a) \le -\epsilon)$ s.t. $\mathbb{P}(|X^k(s,a)| \ge \epsilon) \le \delta/(|\mathcal{S}||\mathcal{A}|(n+1))$. The proof is hence completed by taking the union bound over all $(s,a) \in \mathcal{S} \times \mathcal{A}$ and over all iterations $k$. □

We then derive our analysis under the event specified, i.e.,

$$\|T^k(s,a) - \mathcal{T}_{\mathcal{P}_s^a}(Q^k)\|_\infty \le \epsilon \quad \forall k = 0, 1, \ldots, n, \text{ and } (s,a) \in \mathcal{S} \times \mathcal{A}, \tag{44}$$

which holds with probability $(1-\delta)$ by Proposition D.3.

Moreover, we note that from our R-SAMPLE algorithm, it holds that $\mathbf{Sp}(D^k) \le \mathbf{Sp}(d^k)$. Combining this fact with the nonexpansivity of the $\max$ operator implies that

$$\mathbf{Sp}(T^k - T^{k-1}) = \mathbf{Sp}(D^k) \le \mathbf{Sp}(d^k) = \mathbf{Sp}(h^k - h^{k-1}) \le \mathbf{Sp}(Q^k - Q^{k-1}). \tag{45}$$

We first provide two lemmas.

**Lemma D.4.** *Let $Q^*$ be a solution to the robust Bellman equation $Q^* = \mathcal{T}_\mathcal{P}(Q^*)$. Under the event in equation 44, it holds that*

$$\boldsymbol{Sp}(Q^k - Q^*) \le \boldsymbol{Sp}(Q^0 - Q^*) + \frac{2}{3}\epsilon k, \quad \forall\, k = 0, 1, \ldots, n. \tag{46}$$

*Proof.* By the update rule of RHI, at iteration $k$, it holds that $Q^k = (1-\beta_k)Q^0 + \beta_k T^{k-1}$ with $\beta_k = \frac{k}{k+2}$. We thus have that

$$\mathbf{Sp}(Q^k - Q^*) \le (1-\beta_k)\mathbf{Sp}(Q^0 - Q^*) + \beta_k\mathbf{Sp}(T^{k-1} - Q^*)$$

$$= \frac{2}{k+2}\mathbf{Sp}(Q^0 - Q^*) + \frac{k}{k+2}\mathbf{Sp}(T^{k-1} - Q^*). \tag{47}$$

Using the invariance of $\mathbf{Sp}(\cdot)$ by the addition of constants and the nonexpansivity of $\mathcal{T}_\mathcal{P}$, we can then apply the triangle inequality along with the fact that $\mathbf{Sp}(\cdot) \le 2\|\cdot\|_\infty$ and the bound in equation 44 to obtain,

$$\mathbf{Sp}(T^{k-1} - Q^*) = \mathbf{Sp}\big(T^{k-1} - \mathcal{T}_\mathcal{P}(Q^*)\big) \le 2\epsilon + \mathbf{Sp}(Q^{k-1} - Q^*). \tag{48}$$

We can then plug this back into equation 47 to get

$$\mathbf{Sp}(Q^k - Q^*) \le \frac{2}{k+2}\mathbf{Sp}(Q^0 - Q^*) + \frac{k}{k+2}\big(2\epsilon + \mathbf{Sp}(Q^{k-1} - Q^*)\big).$$

Set $\theta_k = (k+1)(k+2)\mathbf{Sp}(Q^k - Q^*)$, then we have $\theta_k \le \theta_0(k+1) + 2\epsilon k(k+1) + \theta_{k-1}$. Through induction we can get that,

$$\theta_k \le \theta_0 \sum_{i=1}^{k}(i+1) + 2\epsilon \sum_{i=1}^{k} i(i+1) + \theta_0$$

$$= \theta_0 \frac{1}{2}(k+1)(k+2) + \frac{2}{3}\epsilon k(k+1)(k+2).$$

Dividing both sides by $(k+1)(k+2)$ hence completes the proof. □

**Lemma D.5.** *Under the event in equation 44. We denote $\rho_k \triangleq 2\boldsymbol{Sp}(Q^0 - Q^*) + \frac{2}{3}\epsilon k$, then for all $k = 1, 2, \ldots, n$, we have*

$$\boldsymbol{Sp}(Q^k - Q^{k-1}) \le \frac{2}{k(k+1)} \sum_{i=1}^{k} \rho_{i+2}.$$

*Proof.* We have shown two equations:

$$Q^k = \frac{2}{k+2}Q^0 + \frac{k}{k+2}T^{k-1}, \quad Q^{k-1} = \frac{2}{k+1}Q^0 + \frac{k-1}{k+1}T^{k-2}. \tag{49}$$

We then subtract them and have that

$$Q^k - Q^{k-1} = \frac{2}{(k+1)(k+2)}\left(T^{k-1} - Q^0\right) + \frac{k-1}{k+1}\left(T^{k-1} - T^{k-2}\right). \tag{50}$$

By $\mathbf{Sp}(Q^k - Q^*) \le \mathbf{Sp}(Q^0 - Q^*) + \frac{2}{3}\epsilon k$ (from Lemma D.4) and equation 48, we then have that

$$\mathbf{Sp}(T^{k-1} - Q^0) \le \mathbf{Sp}(T^{k-1} - Q^*) + \mathbf{Sp}(Q^* - Q^0)$$
$$\le \rho_{k+2}.$$

Substituting this into equation 50 and using equation 45 yields

$$\mathbf{Sp}(Q^k - Q^{k-1}) \le \frac{2}{(k+1)(k+2)}\rho_{k+2} + \frac{k-1}{k+1}\mathbf{Sp}(Q^{k-1} - Q^{k-2}).$$

We further set $\tilde{\theta}_k = k(k+1)\mathbf{Sp}(Q^k - Q^{k-1})$, and it holds that

$$\tilde{\theta}_k \le \frac{2k}{k+2}\rho_{k+2} + k(k-1)\mathbf{Sp}(Q^{k-1} - Q^{k-2})$$
$$\le \frac{2k}{k+2}\rho_{k+2} + \tilde{\theta}_{k-1}$$
$$\le 2\rho_{k+2} + \tilde{\theta}_{k-1}$$
$$\le 2\sum_{i=1}^{k}\rho_{i+2}.$$

Dividing both sides by $k(k+1)$ implies that

$$\mathbf{Sp}(Q^k - Q^{k-1}) \le \frac{2}{k(k+1)}\sum_{i=1}^{k}\rho_{i+2}, \tag{51}$$

which completes the proof. □

**Theorem D.6** (Restatement of Theorem 4.2)**.** *Consider the exact robust Halpern iteration* $[Q^{k+1}] = [(1-\beta_{k+1})Q^0 + \beta_{k+1}\mathcal{T}_{\mathcal{P}}(Q)]$, *with* $\beta_k = \frac{k}{k+2}$. *Set* $\pi^k$ *to be the greedy policy w.r.t.* $Q^k$. *Then,*

$$\mathbf{Sp}(\mathcal{T}_{\mathcal{P}}(Q^k) - Q^k) \to 0, \text{ and } g_{\mathcal{P}}^* - g^{\pi^k} \to 0, \text{ as } k \to \infty. \tag{52}$$

*Proof.* By Lemma D.1, we have that $g_{\mathcal{P}}^* - g_{\mathcal{P}}^{\pi^k} \le \mathbf{Sp}(\mathcal{T}_{\mathcal{P}}(Q^k) - Q^k)$, thus it suffices to show that $\mathbf{Sp}(\mathcal{T}_{\mathcal{P}}(Q^k) - Q^k) \to 0$.

We derive our analysis under the event in Proposition D.3, that with probability at least $(1-\delta)$, we have that $\|T^k - \mathcal{T}_{\mathcal{P}}(Q^k)\|_\infty \le \epsilon$ for all $(s,a) \in \mathcal{S} \times \mathcal{A}$ and for all $k = 0, 1, \ldots, n$.

For ease of reading, we drop the brackets from the equivalence class notations. Our RHI updates as $Q^k = (1-\beta_k)Q^0 + \beta_k T^{k-1} = \frac{2}{k+2}Q^0 + \frac{k}{k+2}T^{k-1}$ in the quotient space, which implies that for each $(s,a) \in \mathcal{S} \times \mathcal{A}$, we have the following decomposition

$$\mathcal{T}_{\mathcal{P}}(Q^k) - Q^k \tag{53}$$
$$= \frac{2}{k+2}\underbrace{\left(\mathcal{T}_{\mathcal{P}}(Q^k) - Q^0\right)}_{\textbf{Term 1}} + \frac{k}{k+2}\underbrace{\left(\mathcal{T}_{\mathcal{P}}(Q^k) - \mathcal{T}_{\mathcal{P}}(Q^{k-1})\right)}_{\textbf{Term 2}} + \frac{k}{k+2}\underbrace{\left(\mathcal{T}_{\mathcal{P}}(Q^{k-1}) - T^{k-1}\right)}_{\textbf{Term 3}}.$$

We then bound the three terms.

**Term 1:**
Recall that $\rho_k = 2\mathbf{Sp}(Q^0 - Q^*) + \frac{2}{3}\epsilon k$. From the invariance of $\mathbf{Sp}(\cdot)$ by additive constants, the triangle inequality, the nonexpansivity of $\mathcal{T}_\mathcal{P}(\cdot)$ under the span seminorm, and Lemma D.4, it yields

$$\mathbf{Sp}(\mathcal{T}_\mathcal{P}(Q^k) - Q^0) \le \mathbf{Sp}(Q^k - Q^*) + \mathbf{Sp}(Q^* - Q^0)$$
$$\le \rho_k, \quad \forall (s,a) \in \mathcal{S} \times \mathcal{A}.$$

**Term 2:**
This term can be bounded through a similar approach to Lemma D.5 as

$$\mathbf{Sp}\big(\mathcal{T}_\mathcal{P}(Q^k) - \mathcal{T}_\mathcal{P}(Q^{k-1})\big) = \mathbf{Sp}(Q^k - Q^{k-1})$$
$$\le \frac{2}{k(k+1)} \sum_{i=1}^{k} \rho_{i+2}, \quad \forall (s,a) \in \mathcal{S} \times \mathcal{A}.$$

**Term 3:**
From Proposition D.3 we have that

$$\mathbf{Sp}(\mathcal{T}_\mathcal{P}(Q^{k-1}) - T^{k-1}) \le \epsilon, \quad \forall (s,a) \in \mathcal{S} \times \mathcal{A}.$$

We then combine all three terms in equation equation 53, and we have that

$$g_\mathcal{P}^* - g_\mathcal{P}^{\pi^k}(s)$$
$$\overset{Lemma\ 4.1}{\le} \mathbf{Sp}(\mathcal{T}_\mathcal{P}(Q^k) - Q^k) \tag{54}$$
$$\overset{equation\ 53}{\le} \frac{2}{k+2}\rho_k + \frac{k}{k+2}\left[\frac{2}{k(k+1)}\sum_{i=1}^{k}\rho_{i+2}\right] + \frac{k}{k+2}(2\epsilon) \tag{55}$$
$$\overset{(a)}{\le} \frac{4}{k+2}\mathbf{Sp}(Q^0 - Q^*) + \frac{4\epsilon k}{3(k+2)} + \frac{2}{(k+1)(k+2)}\left[\sum_{i=1}^{k}\left(2\|Q^0 - Q^*\|_\infty + \frac{2}{3}\epsilon(i+2)\right)\right] + 2\epsilon \tag{56}$$
$$\overset{(b)}{\le} \frac{4}{k+2}\mathbf{Sp}(Q^0 - Q^*) + \frac{4\epsilon k}{3(k+2)} + \frac{4k}{(k+1)(k+2)}\mathbf{Sp}(Q^0 - Q^*) + \frac{2\epsilon(k^2 + 5k)}{3(k+2)(k+1)} + 2\epsilon \tag{57}$$
$$\le \frac{4(1+2k)}{(k+2)(k+1)}\mathbf{Sp}(Q^0 - Q^*) + \frac{4(3k^2 + 8k + 3)}{3(k+2)(k+1)}\epsilon \tag{58}$$
$$\le \frac{8\mathbf{Sp}(Q^0 - Q^*)}{k+2} + 4\epsilon, \tag{59}$$

where inequality $(a)$ is from the definition of $\rho_k$, inequality $(b)$ is from $\mathbf{Sp}(\cdot) \le 2\|\cdot\|_\infty$.

The proof is thus completed by letting $k \to \infty$ and $\epsilon \to 0$. $\qquad\square$

**Theorem D.7** (Restatement of Theorem 4.3 - Performance of RHI). *Consider a robust AMDP defined by contamination or $\ell_p$-norm, satisfying Assumption 3.1. Set the step sizes $c_k = 5(k+2)\ln^2(k+2)$ and $\beta_k = k/(k+2)$. Then, with probability at least $1 - \delta$, the output policy $\pi^n$ is $\epsilon$-optimal:*

$$g_\mathcal{P}^* - g_\mathcal{P}^{\pi^n}(s) \le \epsilon, \tag{60}$$

*as long as the total iteration number $n$ exceeds $\frac{\mathcal{H}}{\epsilon}$, resulting in the total sample complexity of*

$$\tilde{\mathcal{O}}\left(\frac{SA\mathcal{H}^2}{\epsilon^2}\right). \tag{61}$$

*Proof.* Using the fact that $\mathbf{Sp}(d^k) \leq \mathbf{Sp}(Q^k - Q^{k-1})$ in equation 45 and Lemma D.5, we can derive

$$\mathbf{Sp}(d^k) \leq \frac{2}{k(k+1)} \sum_{i=1}^{k} \rho_{i+2}$$

$$= \frac{4}{k+1} \mathbf{Sp}(Q^0 - Q^*) + \frac{2(k+5)}{3(k+1)} \epsilon$$

$$\leq \frac{4}{k+1} \mathbf{Sp}(Q^0 - Q^*) + 2\epsilon.$$

Since in RHI, in each step $k$, we sample $m_k$ samples for each $(s,a)$-pair, thus the total sample complexity is $SA|\sum_{k=0}^{n} m_k$. Note that

$$m_k = \max\{\lceil \alpha c_k \mathbf{Sp}(d^k)^2/\epsilon^2 \rceil, 1\} \leq 1 + \alpha c_k \mathbf{Sp}(d^k)^2/\epsilon^2, \tag{62}$$

thus we have that

$$\sum_{k=0}^{n} m_k$$

$$\leq (n+1) + \frac{\alpha}{\epsilon^2} \sum_{k=0}^{n} c_k \mathbf{Sp}(d^k)^2$$

$$\leq (n+1) + \frac{10\alpha}{\epsilon^2} \ln^2(2) \mathbf{Sp}(Q^0)^2 + \frac{5\alpha}{\epsilon^2} \sum_{k=1}^{n} (k+2) \ln^2(k+2) \Big( \frac{4\mathbf{Sp}(Q^0 - Q^*)}{k+1} + 2\epsilon \Big)^2$$

$$\leq (n+1) + \frac{10\alpha}{\epsilon^2} \ln^2(2) \mathbf{Sp}(Q^0)^2 + \sum_{k=1}^{n} \frac{240\alpha}{\epsilon^2(k+1)} \ln^2(k+2) \mathbf{Sp}(Q^0 - Q^*)^2 + 40\alpha \sum_{k=1}^{n} (k+2) \ln^2(k+2)$$

$$\leq \mathcal{O}\Big( \alpha \mathbf{Sp}(Q^0)^2/\epsilon^2 + \alpha \ln^3(n+2) \mathbf{Sp}(Q^0 - Q^*)^2/\epsilon^2 + \alpha n^2 \ln^2(n+2) \Big), \tag{63}$$

where the penultimate line uses $(a+b)^2 \leq 2a^2 + 2b^2$ and $\frac{k+2}{k+1} \leq \frac{3}{2}$, and the final equality by integral estimation of the sums. Recalling that $\alpha = \ln(2|\mathcal{S}||\mathcal{A}|(n+1)/\delta)$, $L = \ln\left( \frac{2|\mathcal{S}||\mathcal{A}|(n+1)}{\delta} \right) \log^3(n+2)$, $Q^0 = 0$, and since $n \geq \mathcal{H}/\epsilon$, $\mathbf{Sp}(Q^*) \leq \mathcal{H}$, it holds that

$$SA|\sum_{k=0}^{n} m_k \leq \tilde{O}\big( \frac{SA\mathcal{H}^2}{\epsilon^2} \big), \tag{64}$$

which completes the proof. $\qquad\square$

## E   PF-RHI: A PARAMETER-FREE VARIANT OF RHI

In this section, we present a fully implementable framework for our Robust Halpern Iteration (RHI) algorithm for diverse and unknown problem settings.

As we mention in Remark 4.4, our RHI algorithm does not require any prior knowledge of the underlying robust AMDP, yet the total number of iterations necessary to generate an $\epsilon$-optimal policy is dependent on $\mathcal{H}$. In practice, such an iteration number may need to be pre-set, and it may be infeasible to set for RHI.

In order to bridge this theoretical finite sample complexity result with the nuances of practical application for varying size problem settings, we now extend our RHI algorithm to a more general and implementable framework: PF-RHI, presented in Algorithm 3. Notably, our PF-RHI do not require any knowledge of $\mathcal{H}$ (even the iteration number); and we will show that it finds an $\epsilon$-optimal policy with identical total sample complexity results as RHI, $\tilde{O}\left( \frac{SA\mathcal{H}^2}{\epsilon^2} \right)$.

Note that in our PF-RHI, in each episode $i$, we run the RHI for $n_i$ steps, and output $Q^{n_i}$ and $T^{n_i}$. PF-RHI will terminate if the span $\mathbf{Sp}(T^{n_i} - Q^{n_i})$ is small enough. Hence we do not specify iteration number, and thus no knowledge of $\mathcal{H}$ is needed.

---

**Algorithm 3** Implementable Robust Halpern Iteration (PF-RHI)

---

1: **Input** $Q^0 \in \mathbb{R}^{S \times A}$, $\epsilon > 0$, $\delta \in (0, 1)$, $i = 0$
2: **repeat**
3:     Set $n_i = 2^i$, $\delta_i = \delta/c_i$
4:     Set $\alpha_i = \ln(2|\mathcal{S}||\mathcal{A}|(n_i + 1)/\delta_i)$, $Q^0 = 0$, $T^{-1} = r$, $h^{-1} = 0$, $c_0 = 10 \cdot \ln^2(2)$, $\beta_0 = 0$
5:     **for** $k = 0, \ldots, n_i$ **do**
6:         $c_k = 5(k + 2)\ln^2(k + 2)$, $\beta_k = k/(k + 2)$
7:         $Q^k = (1 - \beta_k) Q^0 + \beta_k T^{k-1}$
8:         $h^k = \max_A(Q^k)$
9:         $d^k = h^k - h^{k-1}$
10:         $m_k = \max\{\lceil \alpha_i c_k \mathbf{Sp}(h^k - h^{k-1})^2/\epsilon^2 \rceil, 1\}$
11:         $D^k = \text{R-SAMPLE}(h^k, h^{k-1}, m_k)$
12:         $T^k = T^{k-1} + D^k$
13:     **end for**
14:     $\pi^{n_i}(s) \in \arg\max_{a \in A} Q^{n_i}(s, a) \quad \forall s \in S$
15:     $i = i + 1$
16: **until** $\mathbf{Sp}(T^{n_i} - Q^{n_i}) \leq 14\epsilon$
17: **Output:** $Q^{n_i}, T^{n_i}, \pi^{n_i}$

---

### E.1 ANALYSIS OF PF-RHI

To facilitate our analysis of PF-RHI, we first present some useful notations as follows.

$$\mu \triangleq \mathbf{Sp}(Q^0 - Q^*),$$
$$\nu \triangleq \mathbf{Sp}(Q^0 - Q^*) + \mathbf{Sp}(Q^0),$$
$$\zeta \triangleq \max\{\mathbf{Sp}(r), \mathbf{Sp}(Q^0)\}.$$

We then define the following random variables:

$$N = \inf\{n_i \in \mathbb{N} : \mathbf{Sp}(T^{n_i} - Q^{n_i}) \leq 14\epsilon\}, \quad \text{and}$$
$$I = \inf\{i \in \mathbb{N} : \mathbf{Sp}(T^{n_i} - Q^{n_i}) \leq 14\epsilon\},$$

and it holds that $N = 2^I$.

We set $i_0 \in \mathbb{N}$ be the smallest integer s.t. $n_{i_0} \geq \mathbf{Sp}(Q^0 - Q^*)/\epsilon = \mu/\epsilon$. Then either $i_0 = 0$ and $n_{i_0} = 1$, or $n_{i_0-1} = n_{i_0}/2 < \mu/\epsilon$, which, when combined, imply that $n_{i_0} \leq 2(1 + \mu/\epsilon)$.

With these, we further define the additional random events:

$$S_i = \{\mathbf{Sp}(T^{n_i} - Q^{n_i}) \leq 14\epsilon, \ \forall(s, a) \in \mathcal{S} \times \mathcal{A}\}, \quad \text{and}$$
$$G_i = \{\|T^k - \mathcal{T}_{\mathcal{P}}(Q^k)\|_\infty \leq \epsilon, \ \forall k = 0, 1, \ldots, n_i, \ \forall(s, a) \in \mathcal{S} \times \mathcal{A}\}$$

where $T^k$ and $Q^k$ are generated by the inner-loop $k = 0, 1, \ldots, n_i$ during the $i$-th iteration of PF-RHI. During this specific iteration $i$, let $M_i$ be the number of samples generated so that $M \triangleq \sum_{i=0}^I M_i$ where $M$ and $M_i$ are random variables.

**Lemma E.1.** *It holds that*

$$\mathbb{P}(S_i) \geq \mathbb{P}(G_i) \geq 1 - \delta_i, \quad \forall i \geq i_0, \forall(s, a) \in \mathcal{S} \times \mathcal{A}. \tag{65}$$

*Proof.* Note that Proposition D.3 directly implies $\mathbb{P}(G_i) \geq 1 - \delta_i$. Moreover, for $i \geq i_0$ and for all $\xi \in G_i$, from Theorem D.6 we have that

$$\mathbf{Sp}\big(T^{n_i}(\xi) - Q^{n_i}(\xi)\big) \leq \mathbf{Sp}\big(T^{n_i}(\xi) - \mathcal{T}_{\mathcal{P}}(Q^{n_i})(\xi)\big) + \mathbf{Sp}\big(\mathcal{T}_{\mathcal{P}}(Q^{n_i})(\xi) - Q^{n_i}(\xi)\big) \tag{66}$$

$$\leq 2\epsilon + \frac{8\mathbf{Sp}(Q^0 - Q^*)}{n_i + 2} + 4\epsilon \tag{67}$$

$$\leq 14\epsilon, \tag{68}$$

thus $G_i \subseteq S_i$, which completes the proof. $\square$

**Proposition E.2.** *It holds that*

$$\mathbb{E}[N] \leq 2(1 + \mu/\epsilon)/(1 - \delta).$$

*Namely, $N$ is finite almost surely and PF-RHI($Q^0, \epsilon, \delta, i = 0$) stops with probability 1 after a finite number of iterations.*

*Proof.* In each iteration $i$, in PF-RHI($Q^0, \epsilon, \delta, i = 0$), we reinitialize $Q^0 = 0$ prior to the inner for loop where $k = 0, 1, \ldots, n_i$. This implies that the events $\{S_i : i \in \mathbb{N}\}$ are mutually independent. Thus,

$$\mathbb{P}(I = i) = \mathbb{P}\Big(\bigcap_{j=0}^{i-1} S_j^c \cap S_i\Big) = \prod_{j=0}^{i-1} \mathbb{P}(S_j^c) \cdot \mathbb{P}(S_i).$$

Now from Lemma E.1, it holds that $\mathbb{P}(S_i^c) \leq \mathbb{P}(G_i^c) \leq \delta_i$ for all $i \geq i_0$, which implies that $\mathbb{P}(I = i) \leq \prod_{j=i_0}^{i-1} \delta_j$.

Moreover, by definition of $c$, we have that $2\sum_{i=0}^{\infty} c_i^{-1} \leq 1$, thus $\delta_j = \delta/c_j \leq \delta/2$, implying that $\mathbb{P}(I = i) \leq (\delta/2)^{i-i_0}$. Using this and the fact that $n_i = n_{i_0} 2^{i-i_0}$, it holds that

$$\mathbb{E}[N] = \sum_{i=0}^{\infty} n_i \mathbb{P}(N = n_i)$$

$$\leq n_{i_0} + \sum_{i=i_0+1}^{\infty} n_{i_0} 2^{i-i_0} \mathbb{P}(I = i)$$

$$\leq n_{i_0}\Big(1 + \sum_{i=i_0+1}^{\infty} \delta^{i-i_0}\Big).$$

The proof is then completed by the bound of $n_{i_0} \leq 2(1 + \mu/\epsilon)$, which implies that $\mathbb{E}[N] \leq 2(1 + \mu/\epsilon)/(1 - \delta)$. $\qquad\square$

**Theorem E.3.** *Let $c_k = 5(k + 2)\ln^2(k + 2)$ and $\beta_k = k/(k + 2)$ hold. Let $n_i = N$ so that $(Q^N, T^N, \pi^N)$ is returned by PF-RHI($Q^0, \epsilon, \delta, i = 0$). Then with probability of at least $(1 - \delta)$, we have for all $s \in \mathcal{S}$,*

$$g_{\mathcal{P}}^* - g_{\mathcal{P}}^{\pi^N}(s) \leq \boldsymbol{Sp}(\mathcal{T}_{\mathcal{P}}(Q^N) - Q^N) \leq 16\epsilon,$$

*Which obtains a sample and time complexity of $\mathcal{O}\big(\hat{L}|\mathcal{S}||\mathcal{A}|(\nu^2/\epsilon^2 + 1)\big)$, with $\hat{L} = \ln\big(4|\mathcal{S}||\mathcal{A}|(1 + \mu/\epsilon)/\delta\big)\log^4(2(1 + \mu/\epsilon))$.*

*Proof.* As Lemma D.1 implies that $0 \leq g_{\mathcal{P}}^* - g_{\mathcal{P}}^{\pi^N} \leq \boldsymbol{Sp}(\mathcal{T}_{\mathcal{P}}(Q^N) - Q^N)$.

We first define $A = \{I \leq i_0\}$ and $B = \bigcap_{i=0}^{\infty} G_i$, where $G_i = \{\|T^k - \mathcal{T}_{\mathcal{P}}(Q^k)\|_\infty \leq \epsilon, \forall k = 0, 1, \ldots, n_i, \forall(s, a) \in \mathcal{S} \times \mathcal{A}$. We claim that, $\mathbb{P}(A \cap B) \geq (1 - \delta)$. To prove this, note that from Proposition D.3, $\mathbb{P}(G_i^c) \leq \delta_i$. Thus

$$\mathbb{P}(B^c) = \mathbb{P}\Big(\bigcup_{i=1}^{\infty} G_i^c\Big) \leq \sum_{i=1}^{\infty} \delta/c_i \leq \delta/2. \tag{69}$$

Then Lemma E.1 implies that $\mathbb{P}(A) \geq \mathbb{P}(S_{i_0}) \geq \mathbb{P}(G_{i_0}) \geq 1 - \delta_{i_0} \geq 1 - \delta/2$, thus combining this with equation 69 implies $\mathbb{P}(A^c \cup B^c) \leq \delta$, which proves our claim.

We then use the definitions of $N$, $I$, and $G_i$, which imply that

$$\boldsymbol{Sp}(\mathcal{T}_{\mathcal{P}}(Q^N) - Q^N) \leq \boldsymbol{Sp}(\mathcal{T}_{\mathcal{P}}(Q^N) - T^N) + \boldsymbol{Sp}(T^N - Q^N)$$
$$\leq 2\epsilon + 14\epsilon$$
$$= 16\epsilon.$$

Applying equation 63 further implies the total sample complexity, $M \triangleq \sum_{i=0}^{I} M_i$ can be bounded as

$$M \leq \sum_{i=0}^{i_0} M_i \tag{70}$$

$$= |\mathcal{S}||\mathcal{A}| \sum_{i=0}^{i_0} \mathcal{O}\Big(\alpha_i \mathbf{Sp}(Q^0)^2/\epsilon^2 + \alpha_i \ln^3(n_i + 2)\mathbf{Sp}(Q^0 - Q^*)^2/\epsilon^2 + \alpha_i n_i^2 \ln^2(n_i + 2)\Big),$$

where $\alpha_i = \ln(2|\mathcal{S}||\mathcal{A}|(n_i + 1)/\delta_i)$ is the parameter defined at iteration $i$ (prior to the inner for loop) of PF-RHI. Moreover, since $n_{i_0}^2 \leq 4(1 + \mu/\epsilon)^2 = \mathcal{O}\big(\mathbf{Sp}(Q^0 - Q^*)^2/\epsilon^2 + 1\big)$, we have that

$$M \leq |\mathcal{S}||\mathcal{A}|(i_0 + 1)\mathcal{O}\big(\alpha_{i_0}\mathbf{Sp}(Q^0)^2/\epsilon^2 + \alpha_{i_0}\log^3(n_{i_0} + 2)\mathbf{Sp}(Q^0 - Q^*)^2/\epsilon^2 + \alpha_{i_0}n_{i_0}^2\log^2(n_{i_0} + 2)\big)$$

$$\leq |\mathcal{S}||\mathcal{A}|\alpha_{i_0}\log^4(n_{i_0} + 2)\,\mathcal{O}\big(\mathbf{Sp}(Q^0)^2/\epsilon^2 + 2\mathbf{Sp}(Q^0 - Q^*)^2/\epsilon^2 + 1\big)$$

$$\leq \hat{L}|\mathcal{S}||\mathcal{A}|\mathcal{O}(\nu^2/\epsilon^2 + 1),$$

which completes the proof. $\qquad\square$

**Corollary E.4.** *Let $n_i = N$. Then with probability of at least $(1 - \delta)$, for all $s \in \mathcal{S}$, it holds that*

$$g_{\mathcal{P}}^* - g_{\mathcal{P}}^{\pi^N}(s) \leq \mathbf{Sp}(\mathcal{T}_{\mathcal{P}}(Q^N) - Q^N) \leq \epsilon.$$

*This results in a sample and time complexity of $\mathcal{O}(\tilde{L}|\mathcal{S}||\mathcal{A}|\mathcal{H}^2/\epsilon^2) = \tilde{\mathcal{O}}(SA\mathcal{H}^2/\epsilon^2)$, where we define $\tilde{L} = \ln(2|\mathcal{S}||\mathcal{A}|\mathcal{H}/(\epsilon\delta))\ln^4(\mathcal{H}/\epsilon)$ and $\tilde{\mathcal{O}}(\cdot)$ hides logarithmic terms.*

*Proof.* Note that

$$\mathbf{Sp}(Q^0 - Q^*) = \mathbf{Sp}(Q^*) = \mathbf{Sp}(r + \mathsf{P}h^*) \leq \mathbf{Sp}(r) + \mathbf{Sp}(h^*),$$

which is due to $Q^0 = 0$ at each iteration $i$ and the nonexpansivity of the map $Q \mapsto \max_{\mathcal{A}}(Q) = h$. Moreover, since $2(1 + \mu/\epsilon) = \mathcal{O}\big(\mathbf{Sp}(h)^2/\epsilon\big)$, combining with Theorem E.3, the result follows by verifying the definition of $\tilde{L}$. $\qquad\square$

We then derive the results under expecations.

**Lemma E.5.** *For an arbitrary fixed iteration $i \in \mathbb{N}$ of PF-RHI$(Q^0, \epsilon, \delta, i = 0)$, let $M_i = |\mathcal{S}||\mathcal{A}| \sum_{j=0}^{n_i} m_j$ be the number of samples obtained during iteration $i$. We have*

$$M_i \leq |\mathcal{S}||\mathcal{A}|\mathcal{O}\big(n_i + (\zeta/\epsilon)^2\alpha_i n_i^2 \log^2(n_i + 2)\big),$$

*where $\alpha_i = \ln(2|\mathcal{S}||\mathcal{A}|(n_i + 1)/\delta_i)$.*

*Proof.* By using induction, for $k = 0$ we have by initialization $d^0 = \max_{\mathcal{A}}(Q^0)$ and $T^{-1} = r$. By using both equation 45 and equation 50 along with the induction hypothesis for $k \geq 0$,

$$\mathbf{Sp}(d^k) \leq \mathbf{Sp}(Q^k - Q^{k-1})$$

$$\leq \frac{2}{(k+1)(k+2)}\mathbf{Sp}(T^{k-1} - Q^0) + \frac{k-1}{k+1}\mathbf{Sp}(d^{k-1})$$

$$\leq \frac{2}{(k+1)(k+2)}\big((k+1)\zeta + \zeta\big) + \frac{k-1}{k+1}\zeta$$

$$= \zeta.$$

This implies that

$$\mathbf{Sp}(T^k) \leq \mathbf{Sp}(T^{k-1}) + \mathbf{Sp}(D^k)$$

$$\leq (k+1)\zeta + \mathbf{Sp}(d^k)$$

$$\leq (k+2)\zeta.$$

Thus for a fixed $i \in \mathbb{N}$ in PF-RHI, we can bound $M_i$ as

$$M_i \leq |\mathcal{S}||\mathcal{A}|\big((n_i + 1) + (\alpha_i/\epsilon^2)\sum_{j=0}^{n_i} c_j \mathbf{Sp}(d^j)^2\big)$$

$$\leq |\mathcal{S}||\mathcal{A}|\big((n_i + 1) + 5(\zeta/\epsilon)^2 \alpha_i \sum_{j=0}^{n_i} (j+2)\ln^2(j+2)\big)$$

$$= |\mathcal{S}||\mathcal{A}|\mathcal{O}\big(n_i + (\zeta/\epsilon)^2 \alpha_i n_i^2 \log^2(n_i + 2)\big),$$

which completes the proof. $\qquad\qquad\qquad\qquad\qquad\qquad\qquad\qquad\qquad\qquad\square$

**Theorem E.6.** *Assume that the robust-AMDP satisfies Assumption 3.1, and that the sequences $c_k = 5(k+2)\ln^2(k+2)$ and $\beta_k = k/(k+2)$ hold. Let $n_i = N$ so that $(Q^N, T^N, \pi^N)$ is the output of PF-RHI$(Q^0, \epsilon, \delta, i = 0)$. Then for every $s \in \mathcal{S}$ we have,*

$$\mathbb{E}\big[g_{\mathcal{P}}^* - g_{\mathcal{P}}^{\pi^N}(s)\big] \leq 16\epsilon + \delta \mathbf{Sp}(r),$$

*which yields an expected sample and time complexity of*

$$\tilde{\mathcal{O}}\big(|\mathcal{S}||\mathcal{A}|(\nu^2/\epsilon^2 + 1 + \delta(1 + \mu/\epsilon)^2(1 + (\zeta/\epsilon)^2)\big).$$

*Proof.* We start our proof similar to Theorem E.3 by considering the events $A = \{I \leq i_0\}$ and $B = \bigcap_{i=1}^\infty G_i$. From Theorem E.3, under $A \cap B$, for every $s \in \mathcal{S}$ it holds that $g_{\mathcal{P}}^* - g_{\mathcal{P}}^{\pi^n}(s) \leq 16\epsilon$ with probability $\mathbb{P}(A \cap B) \geq 1 - \delta$.

On the other hand, under $(A \cap B)^c$, we have the trivial bound of $g_{\mathcal{P}}^* - g_{\mathcal{P}}^{\pi^n}(s) \leq \mathbf{Sp}(r)$, $\forall s \in \mathcal{S}$.

Hence the two cases together imply that

$$\mathbb{E}[g_{\mathcal{P}}^* - g_{\mathcal{P}}^{\pi^n}(s)] \leq 16\epsilon + \delta\mathbf{Sp}(r), \quad \forall s \in \mathcal{S}.$$

Similar to Theorem E.3, we wish to estimate the sample complexity like $M = \sum_{i=0}^I M_i$ for each iteration $i$ of PF-RHI. We accomplish this by considering the infinite disjoint union of all indexes $i > i_0$, or more formally $A^c = \bigsqcup_{i=i_0+1}^\infty \{I = i\}$ which yields

$$\mathbb{E}[M] = \underbrace{\mathbb{E}[M|A \cap B]\mathbb{P}(A \cap B)}_{\textbf{Term 1}} + \underbrace{\mathbb{E}[M|A \cap B^c]\mathbb{P}(A \cap B^c)}_{\textbf{Term 2}} + \underbrace{\sum_{i=i_0+1}^\infty \mathbb{E}[M|I = i]\mathbb{P}(I = i)}_{\textbf{Term 3}}.$$

**Term 1:**
We use the result derived from the proof of Theorem E.3 on the event $(A \cap B)$ and the fact that $\mathbb{P}(A \cap B) \leq 1$. By defining $\hat{L} = \ln\big(4|\mathcal{S}||\mathcal{A}|(1 + \mu/\epsilon)/\delta\big)$, we have that

$$\mathbb{E}[M|A \cap B]\mathbb{P}(A \cap B) = \mathcal{O}\big(\hat{L}|\mathcal{S}||\mathcal{A}|(\nu^2/\epsilon^2 + 1)\big). \tag{71}$$

**Term 2:**
We can combine the result in Lemma E.5 with $\mathbb{P}(A \cap B^c) \leq \mathbb{P}(B^c) \leq \delta$, and $n_{i_0} \leq 2(1 + \mu/\epsilon)$ to obtain the following result:

$$\mathbb{E}[M|A \cap B^c]\mathbb{P}(A \cap B^c) \leq \delta|\mathcal{S}||\mathcal{A}|\sum_{i=0}^{i_0} \mathcal{O}\big(n_i + (\zeta/\epsilon)^2 \alpha_i n_i^2 \log^2(n_i + 2)\big)$$

$$\leq \delta|\mathcal{S}||\mathcal{A}|\mathcal{O}\big(n_{i_0} + (\zeta/\epsilon)^2 \alpha_{i_0} n_{i_0}^2 \log^3(n_{i_0} + 2)\big)$$

$$\leq \delta|\mathcal{S}||\mathcal{A}|\mathcal{O}\big(n_{i_0} + \hat{L}(\zeta/\epsilon)^2 n_{i_0}^2\big). \tag{72}$$

The final inequality holds by using the definition of $\hat{L}$ and that $\alpha_{i_0}\log^3(n_{i_0} + 2) \leq \mathcal{O}(\hat{L})$.

**Term 3:**
To bound this term, we can again employ the result of Lemma E.5 along with defining $Z \triangleq$

$\sum_{i=i_0+1}^{\infty} \mathbb{E}[M|I=i]\mathbb{P}(I=i)$ to have that

$$Z \le |\mathcal{S}||\mathcal{A}| \sum_{i=i_0+1}^{\infty} \mathcal{O}\big(n_i + (\zeta/\epsilon)^2 \alpha_i n_i^2 \log^2(n_i+2)\big)\mathbb{P}(I=i)$$

$$\le |\mathcal{S}||\mathcal{A}| \sum_{i=i_0+1}^{\infty} \mathcal{O}\big(n_i + \hat{L}(\zeta/\epsilon)^2 i^3 n_i^2\big)\mathbb{P}(I=i),$$

where the final inequality follows from using the re-initializations of $n_i$, $\delta_i$, and $\alpha_i$ in PF-RHI to obtain $\alpha_i = \mathcal{O}(\hat{L}+i) \le \hat{L}\mathcal{O}(i)$, where $\log\big((n_i+1)c_i\big) = \mathcal{O}(i)$, and likewise $\log^2(n_i+2) = \mathcal{O}(i^2)$. With this in place, recall that $n_i = n_{i_0} 2^{i-i_0}$. From Proposition E.2, for $i \ge i_0 + 1$ we have that $\mathbb{P}(I=i) \le \prod_{j=i_0}^{i-1} \delta_j \le \mathcal{O}\big(\delta \prod_{j=i_0}^{i-1} \frac{1}{j+2}\big)$. Therefore, we can denote the following

$$S_1 \triangleq \sum_{i=i_0+1}^{\infty} 2^{i-i_0} \prod_{j=i_0}^{i-1} \frac{1}{j+2}, \tag{73}$$

$$S_2 \triangleq \sum_{i=i_0+1}^{\infty} 2^{2(i-i_0)} i^3 \prod_{j=i_0}^{i-1} \frac{1}{j+2}, \tag{74}$$

which allows us to show that

$$Z \le \delta|\mathcal{S}||\mathcal{A}|\mathcal{O}\big(S_1 n_{i_0} + S_2 \hat{L}(\zeta/\epsilon)^2 n_{i_0}^2\big).$$

However, we can calculate equation 73 and equation 74 using their incomplete Gamma functions like,

$$S_1 = e^2 2^{-(i_0+1)}[\Gamma(i_0+2) - \Gamma(i_0+2,2)]$$

$$\le \frac{(e^2-3)}{2} \tag{75}$$

$$S_2 = 84 + 4i_0(i_0+5) + 67e^4 2^{-2(i_0+1)}[\Gamma(i_0+2) - \Gamma(i_0+2,4)]$$

$$= \mathcal{O}\big((i_0+1)^2\big). \tag{76}$$

With equation 75 and equation 76, we can finally bound $Z$ as

$$Z \le \delta|\mathcal{S}||\mathcal{A}|\mathcal{O}\big(n_{i_0} + \hat{L}(\zeta/\epsilon)^2 n_{i_0}^2 (i_0+1)^2\big). \tag{77}$$

We can then find the total expected value of the sample complexity by combining equation 71, equation 72, and equation 77 by rearranging similar order terms and disregarding the logarithmic terms to obtain:

$$\mathbb{E}[M] \le |\mathcal{S}||\mathcal{A}|\mathcal{O}\big(\hat{L}(\nu^2/\epsilon^2+1) + \delta n_{i_0} + \delta\hat{L}(\zeta/\epsilon)^2 n_{i_0}^2 (i_0+1)^2\big)$$

$$= |\mathcal{S}||\mathcal{A}|\tilde{\mathcal{O}}\big((\nu^2/\epsilon^2+1) + \delta(1+\mu/\epsilon)^2(1+(\zeta/\epsilon)^2)\big),$$

which completes the proof. $\qquad\square$

**Corollary E.7.** *Assume that the robust-AMDP satisfies Assumption 3.1, that the sequences $c_k = 5(k+2)\ln^2(k+2)$ and $\beta_k = k/(k+2)$ hold, $r(s,a) \in [0,1]\ \forall(s,a) \in \mathcal{S} \times \mathcal{A}$, and $\mathcal{H} \ge 1$. Let $n_i = N$ such that $N \ge \mathcal{H}/\epsilon$ so that $(Q^N, T^N, \pi^N)$ is returned by PF-RHI$(Q^0, \epsilon/17, \delta, i=0)$ with $Q^0 = 0$, $\epsilon \le 1$, and $\delta = \epsilon^2/17$. We have for every $s \in \mathcal{S}$,*

$$\mathbb{E}[g_{\mathcal{P}}^* - g_{\mathcal{P}}^{\pi^N}(s)] \le \epsilon,$$

*where we obtain an expected sample complexity of $\tilde{\mathcal{O}}\big(|\mathcal{S}||\mathcal{A}|\mathcal{H}^2/\epsilon^2\big)$.*

*Proof.* The proof is directly derived by applying the value of $\delta$ in Theorem E.6. $\qquad\square$

