# OpenReview forum: "Provably Sample-Efficient Robust  Reinforcement Learning with Average Reward"
_ICLR.cc/2026/Conference — ICLR 2026 Conference Withdrawn Submission_

### Official Review · Reviewer_UURo · 2025-10-29

**Soundness:** 3
**Presentation:** 3
**Contribution:** 2
**Rating:** 4
**Confidence:** 3

**Summary:**

The extends optimal sample complexity result to a weaker setting (communicating MDPs) for average reward robust MDPs with uncertainty set bounded by Lp norms, under generative model. It uses RHI to deal with non-existence of contractive Bellman operators in average reward setting.

**Strengths:**

Theorem 4.3 itself is the strength and main contribution of the paper, establishing optimal sample complexity for communicating MDPs.

**Weaknesses:**

As shown in the related work Table 1, similar result already exists for different settings.  I am not sure, how novel and significant contribution it is, to extend to a milder setting of communicating MDPs from already proven settings.

**Questions:**

Q1) Can you please oultline the novelty of the analysis and major challenges faced?

Q2) The paper is limited to generative model. What are the key challenges to  extend this to online settings, like single-time scale as in [1,2]?

Q3) The assumes sa-rectangular robust MDPs which can be very conservative. What are the key challenges to extend this work to s-rectangular or non-rectangular robust MDPs [3,4,5].



[1]@misc{chen2024finitetimeanalysissingletimescaleactorcritic,
      title={Finite-time analysis of single-timescale actor-critic},
      author={Xuyang Chen and Lin Zhao},
      year={2024},
      eprint={2210.09921},
      archivePrefix={arXiv},
      primaryClass={cs.LG},
      url={https://arxiv.org/abs/2210.09921},
}
[2]@inproceedings{
kumar2025on,
title={On the Convergence of Single-Timescale Actor-Critic},
author={Navdeep Kumar and Priyank Agrawal and Giorgia Ramponi and Kfir Yehuda Levy and Shie Mannor},
booktitle={The Thirty-ninth Annual Conference on Neural Information Processing Systems},
year={2025},
url={https://openreview.net/forum?id=OixkI1jSZD}
}

[3]@inproceedings{
kumar2025nonrectangular,
title={Non-rectangular Robust {MDP}s with Normed  Uncertainty Sets},
author={Navdeep Kumar and Adarsh Gupta and Maxence Mohamed ELFATIHI and Giorgia Ramponi and Kfir Yehuda Levy and Shie Mannor},
booktitle={The Thirty-ninth Annual Conference on Neural Information Processing Systems},
year={2025},
url={https://openreview.net/forum?id=Xx0cJGXU7n}
}

[4]@misc{li2025policygradientalgorithmsrobust,
      title={Policy Gradient Algorithms for Robust MDPs with Non-Rectangular Uncertainty Sets},
      author={Mengmeng Li and Daniel Kuhn and Tobias Sutter},
      year={2025},
      eprint={2305.19004},
      archivePrefix={arXiv},
      primaryClass={math.OC},
      url={https://arxiv.org/abs/2305.19004},
}


[5]@misc{gadot2024solvingnonrectangularrewardrobustmdps,
      title={Solving Non-Rectangular Reward-Robust MDPs via Frequency Regularization},
      author={Uri Gadot and Esther Derman and Navdeep Kumar and Maxence Mohamed Elfatihi and Kfir Levy and Shie Mannor},
      year={2024},
      eprint={2309.01107},
      archivePrefix={arXiv},
      primaryClass={cs.LG},
      url={https://arxiv.org/abs/2309.01107},
}

---

### Official Review · Reviewer_g3uJ · 2025-10-31

**Soundness:** 3
**Presentation:** 4
**Contribution:** 3
**Rating:** 8
**Confidence:** 3

**Summary:**

The paper addresses robust reinforcement learning (RL) under the average-reward criterion, which is essential for long-term, steady-state decision-making. This is unlike the discounted-reward setting that overly emphasizes short-term rewards. The key motivation is to bridge the gap between asymptotic results and finite-sample guarantees in robust average-reward RL, especially when the environment is uncertain (due to modeling errors, perturbations, or adversarial noise).

Main Contributions:

1. Theoretical Framework for Robust AMDPs: Introduces analysis of Robust Average Reward MDPs (AMDPs) under a communicating assumption, which is weaker than previous irreducible or ergodic assumptions. It shows that the optimal robust average reward is constant across states. The robust Bellman equation has a solvable form and that its solution defines the optimal robust policy.

2. Algorithm: Robust Halpern Iteration (RHI) It proposes RHI, a model-free and sample-efficient algorithm to find robust average-reward optimal policies. It solves a non-contractive robust Bellman equation using ideas from Halpern iteration, originally from convex optimization. Operates in a quotient space to manage bias shifts and handle the “two-unknowns” problem (policy value and average reward).

3. Finite-Sample Guarantees: Achieves a near-optimal sample complexity.

4. No Prior Knowledge Required of MDP parameters (unlike prior reduction-based approaches). They also present a parameter-free variant  using the "doubling trick" to avoid dependence on unknown constants.

**Strengths:**

1. Strong theoretical contribution: This work makes a foundational advance in robust reinforcement learning (RL) with average reward, a relatively underexplored but important area. It establishes the first finite-sample complexity bound for robust average-reward MDPs (AMDPs), filling a major theoretical gap where prior works offered only asymptotic results. It derives results for communicating MDPs (weaker than irreducibility or ergodicity), and hence significantly broadens the applicability of the theory.

2. Novel algorithmic design:  It introduces Robust Halpern Iteration (RHI), which is a new algorithm inspired by Halpern iteration. HI is a technique in the fixed-point and convex optimization literature, and this work applies it innovatively to robust RL. The algorithm is model-free, data-driven, and does not require prior knowledge of MDP parameters (such as the bias span or transition model). The use of a quotient space formulation to eliminate bias ambiguity is elegant and addresses the two-variable coupling issue in robust Bellman equations.

3. Strong theoretical guarantees: The paper proves near-optimal sample complexity which matches the minimax optimal rate known for non-robust average-reward RL.

4. Broad applicability and generality: The framework accommodates multiple uncertainty models, including the contamination and \ell_p norm models.

5. Empirical Validation: Although primarily theoretical, the experiments are appropriate and well-aligned with the theory.


Overall, this paper provides an important step toward the theoretical maturity of robust average-reward RL and should be of high interest to the machine learning and reinforcement learning theory communities.

**Weaknesses:**

Mild weaknesses:

1.The experimental validation, while consistent with theory, is relatively minimal and performed only on synthetic Garnet environments.

2. The analysis assumes finite state and action spaces, which may limit direct applicability to continuous or high-dimensional settings

**Questions:**

1. In the abstract may not be good to discuss about H since its not introduced yet.

2. Upon reading only the introduction there is a confusion: from the way it is written currently one is not sure whether is the main contribution that of relaxing the irreducibility assumption? of that of extending discounted rewards to average reward? Or is it both. It is both, so please mention this clearly.

3. Line 130: "kernel is not fixed"--very confusing statement, its giving the impression that the MDP is nonstationary. Transition kernel is fixed but can assume any value from the set P.

4. Why is there a somewhat detailed discussion on the discounted case if it is not studied in the current work? Maybe better to move some of that to appendix instead? and use the resulting space to give more intuition on what is the major challenge while going to undiscounted case/ removing irreducibility (some diagrams may help)?

5. line 162: its a function of state s, hence how can one take max until the same policy maximizes for all states? Later on this is proved, but this must be discussed there itself (perhaps footnote).

6.  (9), some intuition regarding utility of this definition of H should be given.

7.  Around line 222: is this true even in the approximate solution case? I.e., if one solves this fixed point equation only approximately, will the resulting greedy policy be near-optimal?

8. Line 232: the term proximal equation can be described more explicitly.

9. Line 238-240: What's the difference between a weak solution and an approximate solution?
Is it simply adding a constant to all elements of the solution vector? How does this help?

10. Sample complexity lower bound? Will help to judge how tight the upper-bound is.

---

### Official Review · Reviewer_R7Hv · 2025-10-31

**Soundness:** 3
**Presentation:** 3
**Contribution:** 3
**Rating:** 6
**Confidence:** 3

**Summary:**

This paper studies the average-reward robust RL problem under the communicating MDP assumption. Two uncertainty sets $\ell_p$-norm and contamination model are considered. The authors apply the Halpern iteration (originally used for non-expansive fixed-point mappings) to the robust Bellman operator iteration, enabling convergence analysis even without contraction property. They further prove a finite-sample complexity bound of $\tilde{O}(SAH^2 / \epsilon^2)$, which is only $\tilde{O}H$-factor worse than the minimax lower bound for standard (non-robust) average-reward MDPs.

**Strengths:**

1. This is the first work to study average-reward robust RL under the communicating MDP assumption with established sample complexity bound.
2. The paper creatively adapts the Halpern iteration to handle the non-expansive robust Bellman operator, ensuring convergence without contraction.

**Weaknesses:**

1. The experiments are limited to synthetic Garnet environments; there are no evaluations on continuous-control or benchmark tasks (e.g., MuJoCo or classic control), leaving the algorithm’s empirical practicality unclear
2. The synthetic transition matrices are randomly generated, but it is not unknown whether they designed transition that is not aperiodic while preserving communication, which is important to demonstrating the claimed generality of the theory. The env with every entry (s,a,s') being positive will not be discriminative across algorithms

**Questions:**

see weakness

---

### Official Review · Reviewer_suFB · 2025-11-03

**Soundness:** 3
**Presentation:** 3
**Contribution:** 2
**Rating:** 4
**Confidence:** 3

**Summary:**

The paper studies distributionally robust average-reward MDPs with only a communicating assumption and proposes Robust Halpern Iteration, a model-free algorithm that solves a quotientspace fixed-point form of the robust Bellman equation. By estimating differences of off-dynamics Bellman operators, the method attains a near-optimal sample complexity $\tilde{O}\left(S A H^2 / \varepsilon^2\right)$ for contamination and $\ell_p$-norm uncertainty sets, without prior parameter knowledge in its analysis.

**Strengths:**

1. The paper works under a communicating robust AMDP assumption (strictly weaker than irreducible/ergodic) and still proves existence of a solution $\left(Q^{\star}, g^{\star}\right)$ to the robust Bellman equation and its equivalence to optimal robust policies, which tightens the theoretical foundation for this setting.

2. Because the robust Bellman operator is only a nonexpansion, the authors import Halpern iteration and run it in a quotient space $\mathbb{R}^{S A} / \sim$, obtaining asymptotic convergence of $\operatorname{Sp}\left(T_P\left(Q_k\right)-Q_k\right) \rightarrow 0$ and thus policy optimality - a nontrivial adaptation of fixed-point methods to robust average reward.

3. Tight, model-free sample complexity: By estimating $T_P\left(Q_k\right)-T_P\left(Q_{k-1}\right)$ under contamination and $\ell_p$-uncertainty and reusing past estimates, the algorithm achieves the currently tightest finitesample guarantee $\tilde{O}\left(S A H^2 / \varepsilon^2\right)$ for robust average-reward RL, and the paper also shows how to remove prior knowledge of $H$ via a doubling trick.

**Weaknesses:**

1. The final rate still scales as $H^2$; since the non-robust minimax rate is $\tilde{O}\left(S A H / \varepsilon^2\right)$, the analysis does not close the extra factor in $H$, and it is left unresolved whether this is intrinsic to robustness or to the proof technique.

2. The recursive estimator R-SAMPLE is tailored to contamination and $\ell_p$-rectangular sets; the method's core variance-reduction step is not shown to work for more general or coupled uncertainty sets.

3. The sample-complexity statement assumes a generative model for the nominal kernel and hides dependence in polylog terms; constants from the non-expansive Halpern recursion and span-based batching are not made explicit, so practical tightness is unclear.

**Questions:**

1. The key reduction "solve $T_P(Q)-Q=c \mathbf{e}$ in the quotient space" hinges on span-invariance: can the authors formalize when this equivalence fails?

2. The Halpern steps $\beta_k=\frac{k}{k+2}$ and the growth of $c_k$ are tuned together with the sampling schedule is there a provably optimal stepsize pair that minimizes the hidden polylog factor in $\tilde{O}(\cdot)$ ?

3. The estimator of $T_P\left(Q_k\right)-T_P\left(Q_{k-1}\right)$ reuses nominal samples to control off-dynamics error; how sensitive is this to inaccurate span estimates $\operatorname{Sp}\left(h_k-h_{k-1}\right)$, which are used to set the batch size $m_k$ ?

4. The result is proved for contamination and $\ell_p$-norm sets; can the same RHI template be extended to Wasserstein or KL balls without losing the $\tilde{O}\left(S A H^2 / \varepsilon^2\right)$ rate?

---

### Official Review · Reviewer_7KMe · 2025-11-06

**Soundness:** 2
**Presentation:** 2
**Contribution:** 2
**Rating:** 2
**Confidence:** 4

**Summary:**

This paper considers a form of Q-learning for robust average-reward MDPs under minimal structural assumptions. Specifically, the MDP is only required to be weakly communicating. Assuming generative access to the nominal model, the authors propose the Robust Halpern Iteration (RHI) algorithm, that handles two uncertainty metrics: the $l_p$ distance and the contamination set. These particular uncertainty sets are chosen because the robust Bellman operator admits an analytical form that depends entirely (and almost linearly) on the nominal transition kernel, making it tractable to estimate from samples. The algorithm outputs an $\epsilon$-suboptimal policy with a sample complexity of $O(SAH²/\epsilon²)$, where $S$ and $A$ denote the state and action space sizes, and $H$ is the optimal bias span.

The theoretical guarantees are validated through simulations showing convergence to the optimal robust average reward. The paper also provides an extensive comparison with prior work across most related domains in RL and MDPs.

**Strengths:**

1. The paper studies a practically important and timely problem which has recently attracted significant attention due to its relevance for long-term decision-making under model uncertainty.
2. It operates under minimal assumptions, requiring only a weakly communicating MDP, whereas most prior works assume stronger conditions such as unichain or ergodicity.
3. The proposed algorithm achieves near-optimal sample complexity, matching the optimal rates for non-robust average-reward MDPs up to a factor of the optimal bias span $H$.

**Weaknesses:**

1. The paper assumes access to a generative model, but this is not made explicit until later in the text. This limits the practical applicability of the results, since trajectory-based data is far more common in real-world settings.
2. The technical novelty is somewhat unclear. Many components of the analysis appear to rely on existing work. For example, Contribution 1 follows almost directly from prior literature, while Contribution 2 builds heavily on ideas from Lee et al. (2025) and the $l_p$ analysis from Kumar et al. (2023). A clearer articulation of the original theoretical advances would strengthen the paper’s contribution.
3. The uncertainty sets ($l_p$ and contamination) are chosen largely because they admit closed-form solutions under the robust Bellman operator, making the analysis tractable. However, this design choice limits the transferability of the approach to other uncertainty metrics that lack such analytical structure.
4. The paper does not provide intuition or a proof sketch for why the proposed algorithm works. Including a high-level explanation of the main technical ideas in the body by potentially by moving the related work to the appendix would make the contributions more accessible.
5. Since the unichain condition does not subsume the weakly communicating condition (the former allows transient states), it is unclear how the stated results extend to unichain settings as claimed in the paper. A clarification of this logical connection would be valuable.
6. Theorem 4.2 analyzes the full-information MDP case and provides only asymptotic convergence, whereas Theorem 4.3 offers finite-sample guarantees for the learning setting. It would be useful to discuss whether finite-time bounds can also be established for the MDP setting in Theorem 4.2.
7. There are concerns regarding certain parts of the proofs, which are elaborated in the following section.

**Questions:**

1. In several parts of the proof (for example, lines 885–886), it is implicitly assumed that the kernel that is worst-case with respect to $h*$
 coincides with the kernel that is worst-case under policy $\pi$. However, these two kernels optimize different objectives where the former minimizes the inner product with $h*$, while the latter minimizes the stationary reward under $\pi$. Could the authors clarify why these two kernels are identical?

2. The evaluation of $\kappa$ in the $l_p$-uncertainty case appears to require knowledge of forbidden states. How is this implemented in practice without access to the underlying model? A clarification on how this term is estimated would be helpful.

Minor Comments:

1. Equation equation appears in many places in the text (for eg: line 208) both in main body and appendix.
2. Line 225, "irreducible or ergodic...." can use some references.

---

### Note · Authors · 2025-11-19

I have read and agree with the venue's withdrawal policy on behalf of myself and my co-authors.